# Multiscale dynamics of charging and plating in graphite electrodes coupling operando microscopy and phase-field modelling

Xuekun Lu [1,2,3] ✉, Marco Lagnoni [4], Antonio Bertei [4], Supratim Das [5], Rhodri E. Owen[1,2], Qi Li [6], Kieran O'Regan [2,7], Aaron Wade [1,2], Donal P. Finegan [8], Emma Kendrick [2,7], Martin Z. Bazant [5,9], Dan J. L. Brett [1,2] & Paul R. Shearing [1,2,10] ✉

The phase separation dynamics in graphitic anodes significantly affects lithium plating propensity, which is the major degradation mechanism that impairs the safety and fast charge capabilities of automotive lithium-ion batteries. In this study, we present comprehensive investigation employing operando high-resolution optical microscopy combined with non-equilibrium thermo-dynamics implemented in a multi-dimensional (1D+1D to 3D) phase-field modeling framework to reveal the rate-dependent spatial dynamics of phase separation and plating in graphite electrodes. Here we visualize and provide mechanistic understanding of the multistage phase separation, plating, inter/intra-particle lithium exchange and plated lithium back-intercalation phenomena. A strong dependence of intra-particle lithiation heterogeneity on the particle size, shape, orientation, surface condition and C-rate at the particle level is observed, which leads to early onset of plating spatially resolved by a 3D image-based phase-field model. Moreover, we highlight the distinct relaxation processes at different state-of-charges (SOCs), wherein thermodynamically unstable graphite particles undergo a drastic intra-particle lithium redistribution and inter-particle lithium exchange at intermediate SOCs, whereas the electrode equilibrates much slower at low and high SOCs. These physics-based insights into the distinct SOC-dependent relaxation efficiency provide new perspective towards developing advanced fast charge protocols to suppress plating and shorten the constant voltage regime.

The large-scale commercialization of electric vehicles (EVs) has been regarded as a critical strategy on the global roadmap to promote a low-carbon economy and minimize global warming. Fast charge capability, driving range and safety of the automotive lithium-ion batteries (LiBs) are the major concerns that influence the broader market uptake of EVs. Graphite anode materials have been prevalently used as the negative electrode in automotive LiBs due to the low and flat working potential, cycling stability, electrolyte compatibility and low cost[1].

However, graphite anode is susceptible to a variety of degradation mechanisms that aggravate battery aging, particularly during fast charging[2–4], restricting the energy and power performance and impairing cell safety.

Metallic lithium deposition on the surface of the graphite anode, also known as lithium plating, is one of the most detrimental degra-dation mechanisms that prevent EVs from charging at a rate that is competitive with refueling a combustion engine vehicle, particularly in

high energy density electrodes where the polarization arising from sluggish mass transport is dominant[5,6]. Plating consumes reversible lithium[7], reduces anode porosity[8] and reaction interfacial area, dendrite formation and propagation, which can lead to internal short circuit[9]. Thus, improved mechanistic understanding of plating propensity associated with electrode microstructure[10], states-of-charge (SOC) and charging speeds is critical to inform the design and optimization of advanced materials and fast charge protocols. This is historically challenging due to the intricate physical and chemical processes across multiple length scales.

Electrochemical techniques such as voltage relaxation[11], differential capacity curves[12], or operando techniques including optical microscopy[13], X-ray/neutron diffraction[14–17] and nuclear magnetic resonance (NMR)[18] are well established to detect lithium plating, however, these methods fail to provide a physics-based understanding of its relationship with the associated microstructure to inform material design. By contrast, multiphysics battery modeling can be used to predict the onset of lithium plating in graphite anodes of different geometries under a variety of operating conditions[19–21]. Unfortunately, most of these models use porous electrode theory[22] and, as a consequence, the impact of microstructural inhomogeneities is neglected, which could exacerbate the non-uniform reaction activity and SOC, leading to an early onset of plating.

Apart from the geometrical simplification, most battery models adopt a Fickian diffusion approach[23] with the gradient of concentration as the driving force to model the solid-state transport in graphite particles[24,25]. However, this cannot capture the characteristic phase-separating behavior in graphite wherein the intercalated lithium tends to separate into Li-rich and Li-poor regions in contrast to an averaged distribution as predicted by the concentration-based Fickian diffusion. This inevitably results in underestimation of the onset and severity of plating. Therefore, the phase-field models based on non-equilibrium thermodynamics[26–28] have become the state-of-the-art modeling method to study phase-separating materials, such as lithium iron phosphate[29] and graphite[30,31]. The solid-state diffusion described by the Cahn-Hilliard style phase-field model is governed by the gradient of chemical potential that is related to the first as well as the third derivative of lithium concentration, so that the uphill diffusion associated with spinodal decomposition can be predicted, which is particularly important to model the mass transport process in graphite particles.

The rational design of fast-charge protocols is equally important as the refinement of electrode microstructure to suppress lithium plating. Although easy to implement, the constant current-constant voltage (CC-CV) method is unsuitable for fast charging due to induced high temperature and plating risk. Thus, many advanced charging protocols are proposed as replacements, such as pulsed current[32,33], varied current[34], multistage constant current[35] or a hybrid profile combining different modes[2,36]. However, designing rational fast charge protocols necessitates fundamental research on the physical processes of lithiation and relaxation at particle and electrode length scales. This is particularly crucial in graphite anodes that exhibit the phase separation (staging) phenomenon; however, the impact of fast charge protocols on the staging phenomena remains to be elucidated. For example, Thomas-Alyea et al.[30] compared the phase-field model with the solid-solution model and found that the gradient of lithium concentration within graphite particles tended to diminish during relaxation if being simulated by the latter, whereas it remained constant in the former due to phase separation. This difference could result in a distinct difference in the design of advanced fast charge protocols, particularly when one or multiple relaxation steps are embedded.

This study aims to underpin the development of next-generation automotive batteries by providing deeper insights into plating mitigation and advanced fast charge protocols through a novel integration of operando optical microscopy and multiscale,

multi-dimensional (1D + 1D to 3D) non-equilibrium phase-field modeling techniques, which appears to be the first implementation in this area. The distinct physical processes of phase separation and plating/stripping are concurrently visualized at different charging currents, which is compared with a phase-field model of graphite intercalation to understand the competing spatial dynamics of intercalation and plating/stripping at the electrode scale. The dependence of intercalation wave/shrinking core mechanisms as a function of particle size, orientation, shape, surface condition and charging rate is experimentally observed and successfully predicted by high-fidelity 3D microstructural-resolved phase-field modeling. Its impact on the heterogeneous reaction and early onset of plating is elucidated. Finally, new insights into the development of the fast charge protocols are provided based on the rate-dependent relaxation dynamics at the electrode and particle scale.

## Results

### Phase separation and plating/stripping by operando optical microscopy

The experimental setup of the operando optical microscopy is shown in Fig. 1. A strip of the graphite working electrode ($2.2 \times 8$ mm$^2$, areal capacity 2 mAh cm$^{-2}$) assembled in an ECC-Opto-Std optical cell (Fig. 1a, EL-cell, Germany) is examined from the top view using the Keyence VHX-7000 optical microscope (Fig. 1b). The top surface of the electrode strip is in close contact with the glass (Fig. 1a), where the electrolyte accessibility is limited, and the bottom of the strip is copper current collector, in contact with the separator. Thus, the Li$^+$ ions mainly intercalate from the lateral surfaces toward deeper region of the graphite electrode (Fig. 1c, d), and accordingly the current density is calculated as the ratio of the total current and the four lateral surface areas of the rectangular graphite electrode. Note that the "thickness" in the optical experiment refers to the in-plane horizontal distance in lithiation direction of the electrode strip (x-direction, Fig. 1c), which is much larger than the conventional thickness of the electrode coating (z-direction, Fig. 1d). This is particularly conducive to capturing the spatial dynamics of lithiation, non-equilibrium phase separation, relaxation and plating/stripping process under the condition of large electrolyte concentration gradient.

Graphite is a well-known phase-separating material exhibiting characteristic optical colors as a function of the SOC[13,21,30] (Fig. 1e) due to a shift in plasma frequency. This makes operando optical microscopy an advantageous technique in revealing the phase evolution in a single graphite particle[21], thick electrode[13] and at the electrode/electrolyte interface[37]. Note that we will use "charge" to represent lithiation of the graphite electrode (i.e., potential of graphite vs. Li/Li$^+$ decreases) to be intuitive throughout this study; the percentage and decimal number of SOC represent the lithium content in the entire electrode (i.e., global SOC) and in the particle, respectively. The electrochemical cycling profile is shown in Fig. 2, comprising charge at 2 mA cm$^{-2}$ (pink) followed by 3 h relaxation (green), discharge (light blue) and another 3 h relaxation (green); the same cycle is repeated next at 4 mA cm$^{-2}$. A current density of 2 mA cm$^{-2}$ corresponds to the current density of 1C in the coin cell experiment presented later on in this study. Despite the different geometries, we will provide evidence later to prove that the onset of plating occurs at identical global SOCs as a consequence of local particle saturation. Moreover, the spatial dynamics of intraparticle redistribution and inter-particle exchange of intercalated lithium is identical between the optical cells and the coin cells.

Figure 2 displays the corresponding optical observations of the electrode surface at specific time steps indicated by the numbers. At SOC = 0%, the graphite particles display the original color of dark gray. After being charged at 2 mA cm$^{-2}$ for 4 h, phase separation at electrode level is observed, indicated by three distinct colors: red (Stage 2, SOC ≈ 0.52), dark blue (Stage 3, SOC ≈ 0.23) and dark gray (Stage 1L, SOC < 0.05). Lithiation is observed to start from the lateral surface of

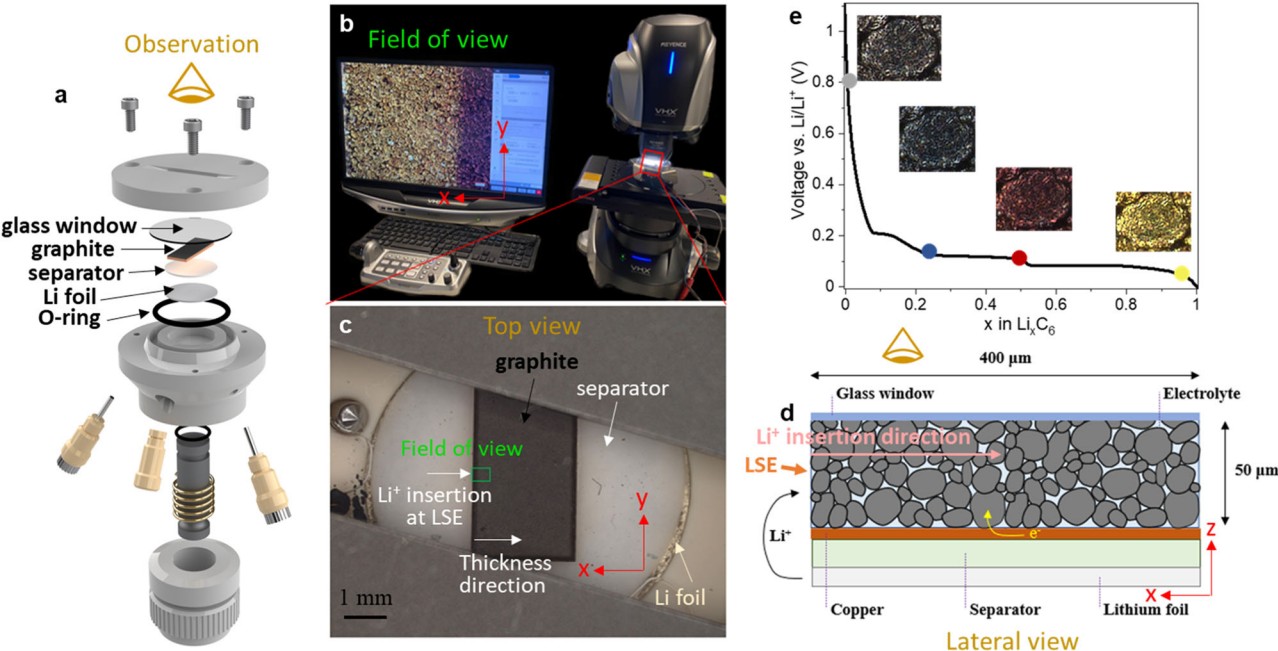

**Fig. 1 | The experimental setup of the operando optical microscopy. a** The optical cell assembly; **b** the experimental setup; **c** the top view of the cell; **d** the schematic of the lateral view of the cell geometry; **e** the SOC vs. optical color of a graphite particle. LSE lateral surface of the electrode.

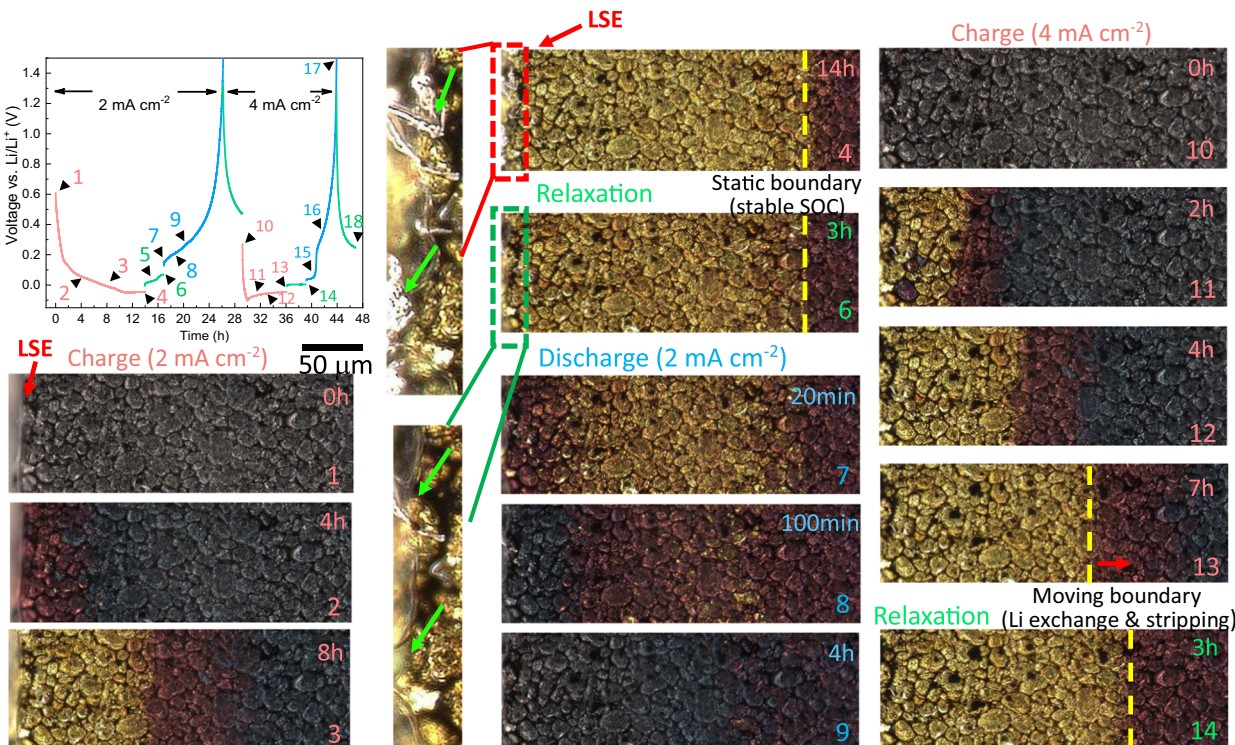

**Fig. 2 | Operando visual investigation of the lithiation and plating/stripping process by optical cell at different cycling currents.** The timestep numbers stamped on the optical images correspond to the ones in the voltage profile. The field-of-view of the optical microscopy shows the first 400 µm of a total thickness of 2.2 mm (i.e., in-plane horizontal direction). Source data are provided as a Source Data file.

the electrode (LSE, Fig. 2), due to the liquid transport resistance in the thickness (lithiation) direction. As the charging proceeds (8 h), a gold region (Stage 1, SOC ≈ 0.95) nucleates and propagates deeper into the electrode along with other colored regions. Three-stage coexistence is seen. Note that the voltage profile becomes negative and reaches a plateau after 10-h charging, indicating the onset of plating, which is

also visually confirmed at the LSE at Timestep 4 (indicated by green arrows).

A relaxation step started immediately after completion of charging (13 h). Notably, the gold region did not shrink immediately but, instead, it propagated slightly within the first 1.5 h (Timestep 5, shown in Fig. S1 in the SI), followed by a color fade and mild shrinkage of the

boundary (Timestep 6). The beginning of this two-step process is believed to be the "back-intercalation" of lithium as a consequence of stripping of the plated lithium, and subsequently the equilibration of the lithium content in graphite. This is not only evidenced by the double-plateau shown in the voltage profile between Timestep 4 and 6, but also by a drastic decrease at the LSE of the visible plated lithium compared to Timestep 4 (see the magnified insets of Timesteps 4 and 6).

The graphite electrode was set to delithiate (discharge) after 3-h relaxation. The gold region decays immediately at the LSE. This is consistent with the charging process, proving that the sluggish electrolyte transport leads to large polarization due to the concentration and potential gradient along the thickness direction, and thus the reaction is more favorable at the LSE due to the faster kinetics. The green arrow in Timestep 6 identifies the remaining plated lithium that has not dissolved over the relaxation period. As the discharge continues, most of the gold region disappears, and the red region starts to fade from the LSE until the end of discharge, when the color of the graphite restores to gray. Note that the residual plated lithium largely disappears during the discharge process.

To examine the rate-dependence of the observed physical processes, the cell was subsequently cycled at $4\,mA\,cm^{-2}$ from SOC = 0 (Timestep 10). A distinct difference is seen between $4\,mA\,cm^{-2}$ (Timestep 11) and $2\,mA\,cm^{-2}$ (Timestep 2), with the former exhibiting a region with much "higher" SOC than the latter, judging by the optical color, although both of them are at the same global SOC. This implies that an intra-particle lithium gradient establishes under the $4\,mA\,cm^{-2}$ charging condition, suggesting the dominant resistance here is solid-state diffusion. As the charge proceeds, the lithiation discrepancy decreases (Timestep 12 vs. Timestep 3) and eventually displays an overall under-lithiated state (Timestep 13) compared to $2\,mA\,cm^{-2}$ (Timestep 4), implying that only a fraction of the charging current drives intercalation of lithium into the graphite. The voltage profile suggests that the electrode was merely charged at the current density of $4\,mA\,cm^{-2}$ for 1 h, beyond which large-scale plating occurred, evidenced by a long plateau after the voltage dip (energy barrier of plating nucleation[21]). As the electrode expanded, the LSE is out of the field of view, but examination after two cycles confirms the existence of irreversible plated lithium at the LSE (see Fig. S2 in the SI). In contrast to what was observed at $2\,mA\,cm^{-2}$, a relaxation step following the $4\,mA\,cm^{-2}$ charge shows continuous propagation of the gold and red regions over the next 3 h, which is believed to arise from the inter-particle lithium exchange as a consequence of the deintercalation of unstable lithium at the saturated graphite surface, and an extensive lithium stripping, substantiated by a voltage plateau at 0 V during the relaxation. The growth of the gold region continues even after the start of discharge, but at a slower speed (see Fig. S1 in the SI for Timesteps 15–18). A video showing the whole process of the operando optical microscopy is uploaded with the paper (Supplementary Movie 1).

In summary, three-stage coexistence is observed in the thick electrode due to the electrolyte transport resistance. The occurrence of plating reduces the effective intercalation current. The stage distribution and phase boundaries remain static during relaxation after low current charging; in contrast, the phase boundaries continue to propagate along the thickness direction during relaxation after high current charging due to inter-particle lithium exchange and lithium stripping, which will be substantiated by phase-field modeling in the next section.

### The dynamics of phase separation and plating/stripping by macroscopic phase-field modeling

To improve the mechanistic understanding of the spatial dynamics of the intercalation and plating/stripping phenomenon observed from the in operando optical experiment, macroscopic phase-field modeling is employed. Geometrical parameters (e.g., tortuosity, graphite phase fraction and porosity, etc.) are obtained from the 3D microstructural characterization shown in the next section. Further details of the model setup and input parameters are presented in the SI. The simulated voltage curves during charge and relaxation are in good agreement with the experiment (Fig. S3), particularly the two-stage relaxation process at low current. It is widely accepted that plating could occur when the anode potential falls below 0 V, however, Gao et al. reported a nucleation energy barrier of plating between 7 to 150 mV[21], which depends on the surface condition such as defects[38,39]. In this study, this value is determined to be 20 mV by matching the predicted phase and SOC distributions (indicated by the optical color) of Li intercalation along the horizontal thickness direction with the operando observation at the end of charge and relaxation for different charging current densities (Fig. S4). This nucleation energy barrier of plating is implemented as part of the plating kinetics applied to individual graphite particles (see the plating kinetics in the SI). More explanation can be found in Supplementary Discussion 1 in the SI. A detailed mechanistic interpretation of the phase-separation governed SOC evolution during charge and relaxation at low and high current, plating and stripping thickness and the impact of porosity assisted by the macroscopic phase field model are presented in the SI (Figs. S4 and S5 and Supplementary Discussion 1).

Figure 3a–e overlays the predicted (de)lithiation current density (blue) and plating/stripping current density (green) with the optical observation. The three peaks in Fig. 3a (from left to right) represent the lithiation current density at the LSE, phase boundary of Stage 2/3 and Stage 3/1L respectively. The intercalation front starts at the LSE and global phase boundary at low SOC, and propagates with time to the deeper area of the electrode (Fig. 3b), aligned with the phase boundary; in the meantime the reaction activity drops significantly at the LSE due to the surface saturation. The intercalation current peaks at the global phase boundary between the two energetically stable neighboring regions. After 10 h lithiation, plating current occurs at the LSE, where the lithiation current reaches zero (Fig. 3c), attributed to a lower chemical potential of the plated lithium than the intercalated lithium, and therefore the plating becomes favorable. After 14 h lithiation, the stripping process starts first during relaxation, with the stripped lithium dissolving into the electrolyte and back-intercalating at the global phase boundaries (Fig. 3d), where the particles are less stable. Meanwhile, the rest of the graphite particles remain static, suggested by the zero (de)lithiation current density. As the process continues, the reversible plated lithium finishes stripping, and intercalated lithium starts to equilibrate, in terms of intra-particle redistribution to minimize the phase boundary and inter-particle exchange to restore to a stable SOC (0.95) at the particle surface; this is evidenced by a gentle fade of the gold color (Fig. 3e). Similar to the lithiation process, the delithiation current density peaks at the global phase boundary. Figure S5b compares the thickness of the plated lithium for the 2 and $4\,mA\,cm^{-2}$ charging at the same SOCs. A pronounced difference in the onset and severity of plating is seen between the two cases. The inset of plating thickness vs. relaxation time shows that at least 3 h are required to fully strip a $0.65\,\mu m$-thick plated layer.

Combining the operando experiment and macroscopic phase field model, it is found that, from the LSE and along the thickness, graphite particles spatially follow the order: saturated Stage 1 (metastable)−Stage 1/2 (stable)−phase boundary (unstable)−Stage 2/3 (stable)−phase boundary (unstable)−Stage 3 during fast charge. The lithiation front initiates at the LSE and then propagates and peaks at phase boundaries, where the most energetically unstable regions are. During relaxation, the stripping process takes place first, with the dissolved lithium back-intercalating at the phase boundary; the exchange current for lithium equilibration (i.e., internal current) increases with time and peaks at the phase boundary.

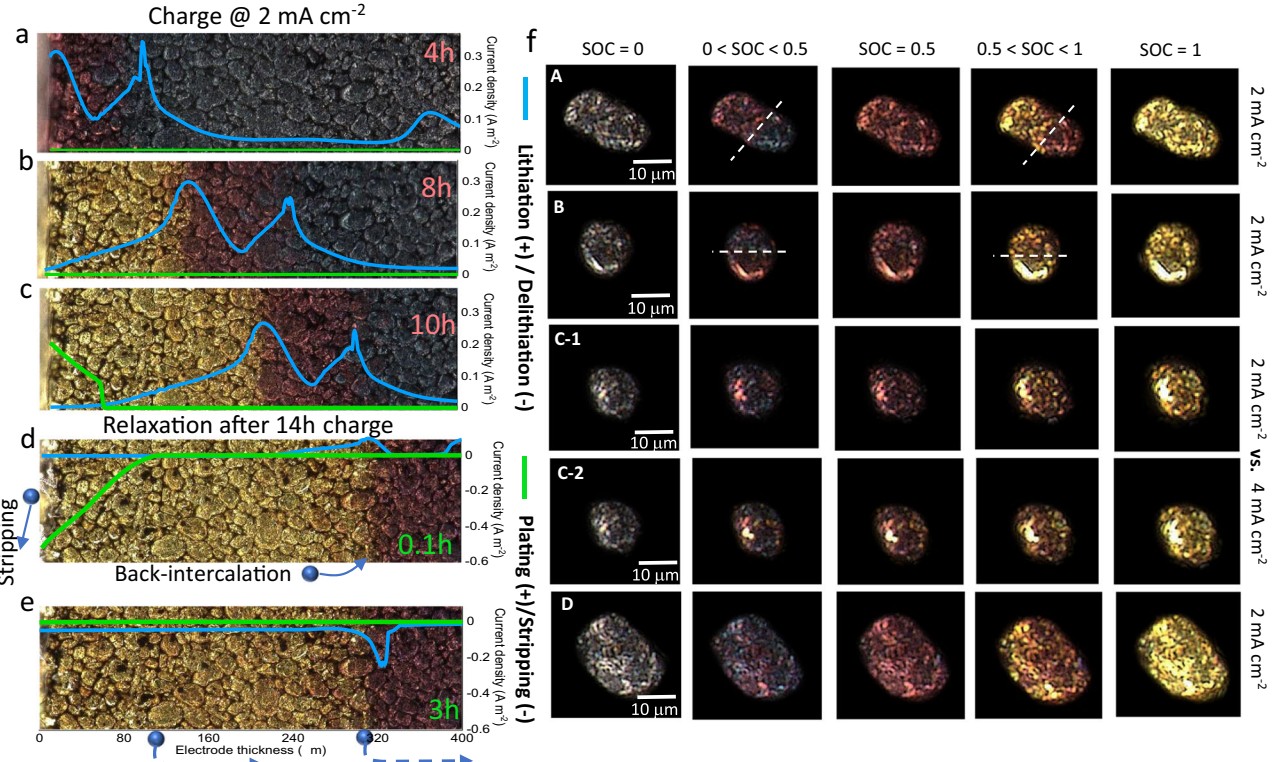

**Fig. 3 | Multiscale dynamics of graphite electrode during charge. a–e** Predicted current density distribution of the lithiation/delithiation (blue profiles) and plating/stripping (green profiles) reactions during charge and relaxation by macroscopic phase-field modeling; **f** rate-dependent phase separation and heterogeneities at single particle level observed within the field of view. Particles A, B, C and D represent the typical lithiation mechanisms observed in the graphite electrode, extracted from the locations as shown in Fig. S6a.

Whilst macroscopic analysis reveals the spatial dynamics of lithiation/plating propensity and heterogeneity at the electrode length scale, it is also crucial to understand the physical process and inhomogeneities at the particle level, which can provide additional insights into the morphological design to mitigate local onset of plating during fast charge. Figure 3f compares the lithiation behavior in four different particles extracted from a similar location (Fig. S6a), representing the distinct phase separation mechanisms observed in the field of view, depending on the shape, orientation, surface condition and size of graphite particle and charging current. Particle A displays an elliptical geometry with the long axis aligned 45° to the global lithiation direction. The particle has a well-defined phase boundary and the lithiation follows an "intercalation wave" manner ($0 < SOC < 0.5$). The particle shape is noted to affect the intercalation behavior since the phase boundary propagates along the longitudinal direction of the particle rather than in the global lithiation direction. The phase boundary sweeps across the particle with time until reaching $SOC = 0.5$, where a uniform color distribution is observed, meaning the end of Stage 3/2 coexistence. As charging continues, Stage 1 (gold) starts to nucleate and propagate from the left just like Stage 2, and the particle finally becomes fully gold ($SOC = 1$).

In contrast, Particle B is less elongated, with a noticeable bright edge at the bottom, where the intercalation onsets, which then sweeps toward the top. This bright edge might be associated with exposed graphene layers acting as a highway for intercalation and the solid-state transport, or it could just be a sharp-edge inducing strong light reflection and the preferential lithiation at the edge is due to its high area to volume ratio (i.e., fast kinetics). Phase boundary movement along the longitudinal direction in both Particle A and B could be explained from the perspective of energy minimization. A shorter length of the phase boundary is more favorable since it is highly energetically unstable. In contrast, Particle C shows no internal phase boundary in the Stage 2/3 ($SOC < 0.5$) nor Stage 1/2 ($0.5 < SOC < 1$) regime; lithiation nucleates at multiple sites and spreads to cover the full surface. Unlike Particle B, the bright spots in Particle C are disconnected and localized (i.e., 0D entrance), implying that the surface grains have distinct orientations, and therefore it is more difficult to initiate the nucleation of the intercalation wave. When the lithiation current is doubled ($4\,mA\,cm^{-2}$), it introduces a marked intra-particle SOC heterogeneity (Particle C-2). Particularly, three-stage (gold/red/dark blue) coexistence is observed even when the SOC is less than 0.5, which is highly related to the morphological heterogeneity at the particle surface. Apart from the surface condition, particle size is another major factor that determines the intercalation mechanism, as the time constant for solid-state diffusion squares with the radius. Particle D with a diameter of ~20 μm exhibits a typical "shrinking core" behavior, as the solid-state transport is limiting[40]. Figure S6b plots the relative rate between solid-state diffusion and reaction kinetics at $2\,mA\,cm^{-2}$, from which the transition between intercalation wave (kinetic dominance) and shrinking core (solid-state transport dominance) is observed to be ~7 μm.

In summary, the lithiation mechanism, either by intercalation wave or shrinking core, is dependent on particle shape, orientation, surface condition, particle size and charging current. These significantly determine the spatial plating heterogeneities, both in the depth and in-plane directions and at the particle level. However, intra and inter-particle SOC heterogeneities cannot be captured by macroscopic modeling techniques and yet are the major causes of the early onset of lithium plating. Thus, a microstructure-resolved method, with the particle shape, orientation, surface conditions (sharp corners and edges) included, is needed to provide deeper insights into this issue, which is explored in the next section.

## 3D microstructure characterization

To correlate the spatial dynamics of phase separation and plating with the architecture at the particle and electrode level, the 3D morphology of a 2 mAh cm$^{-2}$ graphite electrode is obtained using X-ray CT and visualized in Fig. 4a. The volume fraction of graphite particles is measured to be 0.65. The spatial pore size distribution is shown in Fig. 4b, wherein a heterogeneous and wide size distribution is found. Figure 4c shows a Li$^+$ flux map simulated based on the pore structure in Fig. 4b using an image-based modeling approach[41]. The pore pathways with bright colors indicate a local pore bottleneck and therefore high electrolyte transport resistance. Based on the diffusive flow result, a much higher tortuosity factor is found in the vertical (along the thickness, $\tau_V = 7.3$) direction than the horizontal ($\tau_H = 3.87$) direction, which could be attributed to the orientation and shape of the graphite particles and the anisotropic pore structure after calendering. Figure 4d displays the pore phase with the color coding showing the geometric tortuosity, which is lower than $\tau_V$ (6 vs. 7.3), highlighting the extra resistance arising from the constriction effect of the pore throat on diffusive flux in Fig. 4c. Depending on the orientation, all the pores can be classified into horizontal and vertical groups (Fig. 4e). The former takes up to 58% of the total pore phase ($\varepsilon = 0.28$) against the latter (42%). This is believed to be a consequence of the calendering process that drives most of the elliptical graphite particles to align in the horizontal direction, as shown in Fig. 4f, wherein a wide particle size distribution (color-coded) is observed. Figure 4g, h visualizes the size distribution of the vertical and horizontal pore respectively, and their slice-by-slice variation is plotted in Fig. 4i. It is inferred that the horizontal pore is more susceptible to the calendering due to the reduced pore size, although the tortuosity is higher in the vertical direction due to the lower porosity and percolation. Figure 4j shows that most of the graphite particles have their long axis aligned between 60° to 90° relative to the global lithiation direction, but there is no

preferential arrangement in the horizontal plane. The impact of particle size, orientation, pore morphology and distribution on the lithiation and plating behavior will be discussed later.

## Rate-dependent early plating detected by OCV relaxation profiles

The early onset of plating is detected experimentally by identifying the inflection point of the open circuit voltage (OCV) profile during relaxation. The cycling profile is shown in Fig. 5a. The cell is charged incrementally at a step size of 5% SOC, followed by the OCV profile examination during rest, after which the cell is slowly discharged back to 0% SOC. Figure 5b shows the OCV curves of a coin cell (mass loading 1 mAh cm$^{-2}$) charged at 1C. It is noted that as the SOC increases, the voltage plateau decreases. Moreover, it takes less than 5 min to reach equilibrium at 35 and 40% SOC but takes longer at 45 and 50% SOC, which is related to the accumulation of unstable intermediate SOCs due to the non-equilibrium lithiation and shrinking core mechanism, particularly toward the end of the Stage 3/Stage 2 plateau (see Fig. S5 and SI Discussion). Relaxation at this SOC includes not only the redistribution of the lithium content and minimization of the phase boundary, but also the delithiation of the unstable intermediate phase that re-intercalates at the deeper region of the electrode where the SOC is lower, known as the galvanic corrosion process[37]. A rapid relaxation occurs above 55% SOC due to the nucleation of Stage 1. The same trend is also found in the 2 mAh cm$^{-2}$ cell.

It is noted that an inflection appears when the SOC exceeds 80% in the 1 mAh cm$^{-2}$ cell, but this occurs much earlier (SOC = 65%) in the 2 mAh cm$^{-2}$ cell (Fig. 5c). The inflection of the OCV curve is known to be an indication of lithium plating[42,43]. The curve is divided into two plateaus, the first of which corresponds to the stripping process and the dissolved lithium back-intercalating in a deeper area of the electrode to maintain neutrality, thus slowing down the rise of the OCV curve and

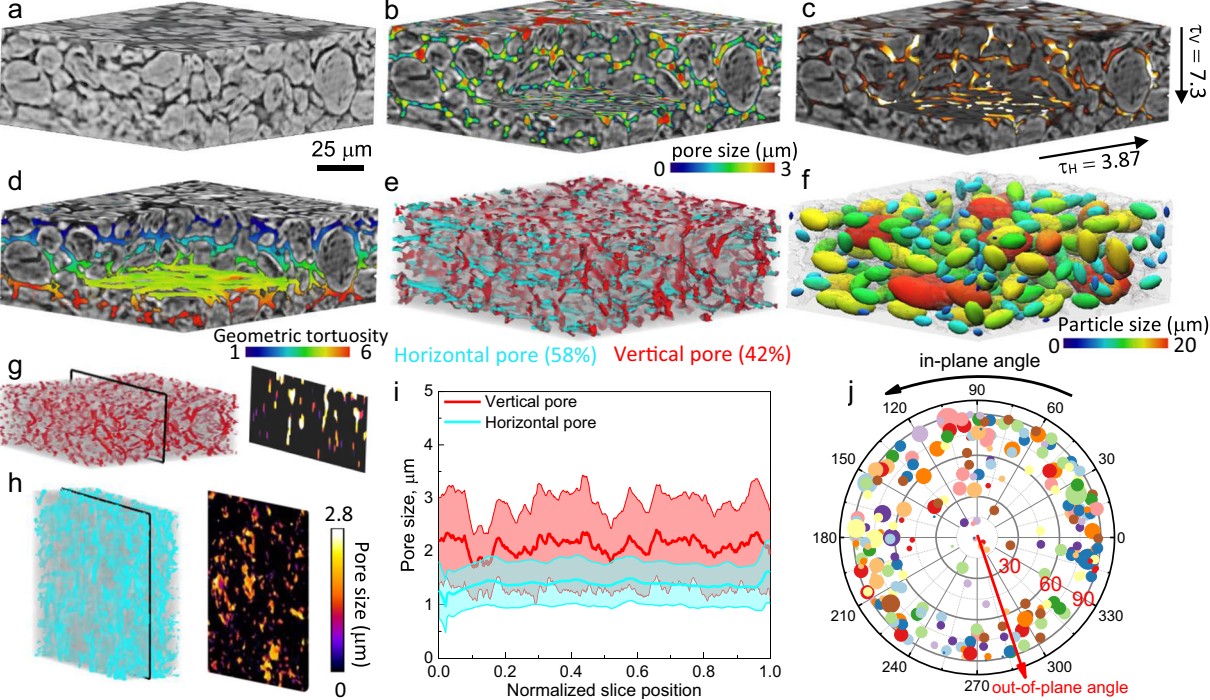

**Fig. 4 | 3D microstructure characterization of a 2 mAh cm$^{-2}$ graphite electrode. a** 3D volume rendering of the electrode; **b** visualization of the pore phase with the color code showing the pore size distribution; **c** diffusive flux obtained by CFD simulation based on the pore structure in (**b**); **d** spatial distribution of the geometric tortuosity; **e** visualization of the horizontal and vertical pore; **f** graphite particles represented by elliptical templates with the color-code showing the size distribution; **g, h** comparison of the size distribution of the horizontal and vertical pore on a cross-sectional slice; **i** quantitative comparison of the size distribution of horizontal and vertical pore for all the slices; **j** statistical analysis of the in-plane and out-of-plane orientation of the particles. The symbol size scales with the particle size and the symbol colors are randomly assigned for the ease of visualization. Source data are provided as a Source Data file.

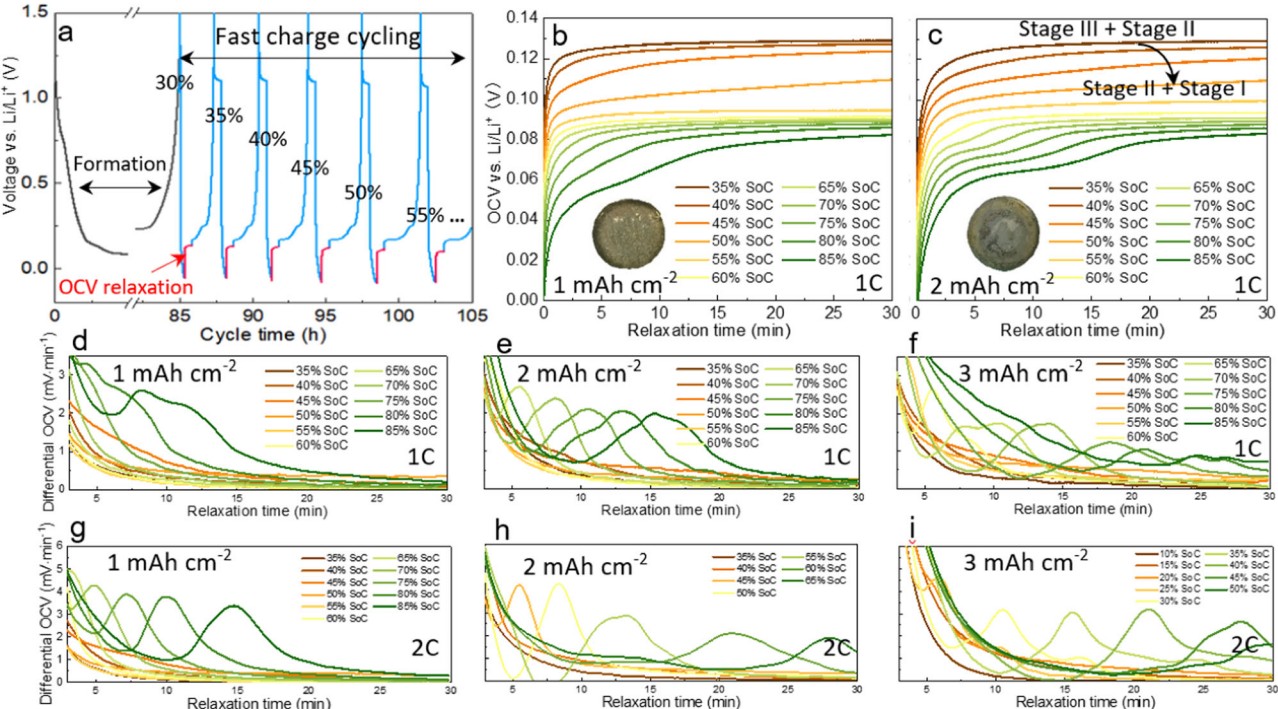

**Fig. 5 | Detection of the early onset of plating by the OCV relaxation method. a** The experimental cycling profiles. The measured OCV curves during relaxation after being charged to each SOC level at 1C in a (**b**) 1 mAh cm$^{-2}$ and (**c**) 2 mAh cm$^{-2}$ cell; **d–f** differential OCV vs. relaxation time under 1C charge is plotted for three mass loading cells; **g–i** differential OCV vs. relaxation time under 2C charge is plotted for three mass loading cells. Insets in (**b**) and (**c**) show the optical image of the electrode surface disassembled from the coin cell after being lithiated to 85% SOC. Source data are provided as a Source Data file.

delaying the equilibration. The thicker electrode shows a more predominant inflection due to more severe plating. The physical process behind the inflection point and first plateau was previously captured by Chen et al.[37], who observed the stripping of plated lithium that back-intercalated into the low SOC regions, accompanied by formation of dead lithium detached from the electrode as a consequence of the volume contraction of the plated lithium. Differential OCV vs. relaxation time is plotted for 1, 2 and 3 mAh cm$^{-2}$ cells charged at 1C (Fig. 5d–f) and 2C (Fig. 5g–i) respectively, enabling the identification of the inflection points of the OCV curve as a peak in the differential OCV curve. It is straightforward to see the first inflection point (peak) appears earlier for 2C charging and thicker electrodes, e.g., 80% SOC (1C) vs. 70% SOC (2C) for 1 mAh cm$^{-2}$, 65% SOC (1C) vs. 40% SOC (2C) for 2 mAh cm$^{-2}$ and 60% SOC (1C) vs. 25% SOC (2C) for 3 mAh cm$^{-2}$. Moreover, the peaks shift toward the right side as the C-rate and thickness increase, indicating a prolonged stripping/back-intercalation process due to the increased lithium plating, thereby requiring a longer relaxation period. This is consistent with the earlier observations[37].

The electrochemical method clearly demonstrates a strong dependence of the plating and stripping behavior on charge rate and electrode thickness[44]; however, the voltage response alone is unable to reveal the plating/stripping dynamics at the microscale and their interplay with the electrode morphology. This highlights the impetus for developing the 3D microstructure-resolved phase-field modeling to provide a physics-based understanding of the plating propensity and the SOC-dependent relaxation behavior to assist the rational design of fast charge protocols.

## Impact of 3D microstructure on phase separation and lithium plating

Compared with the 2D homogenized modeling, 3D microstructure-resolved modeling provides additional insights into the influence of microstructural heterogeneities on the physical processes and battery

performance, since (1) no volume-averaged parameters such as porosity or particle size distribution are used to simplify the electrode geometry[45]; (2) the particle morphology (i.e., shape, edges and corners) is genuinely represented (Fig. S7). This is particularly important in studying the early plating phenomenon. Figure 6a, b compares the distinct lithium content distribution alongside the voltage curves simulated at 0.05C (0.02 mA cm$^{-2}$) and 1C (0.4 mA cm$^{-2}$) respectively in a single graphite particle. At low C-rate, the phase boundary is transversal to the particle following the same intercalation-wave mechanism as observed experimentally (Particle A in Fig. 3f). A relaxation step following the charging shows little change in the OCV profile and lithium distribution, suggesting an equilibrated charging process. In contrast, a distinct difference is observed when charging at 1C. Lithiation governed by the shrinking core mechanism induces a gradient from shell to core of the particle (Fig. S8), consistent with the experimental observation of Particles C and D (Fig. 3f). This inevitably results in a SOC saturation at the outer surface and underutilization of the capacity toward the core of the particle. This non-equilibrium charge leads to a drastic response of the OCV profile at rest, including a sharp increase at the beginning of the relaxation due to the ohmic loss, followed by a gentler voltage rise attributed to lithium redistribution to minimize the configuration energy from an unstable state, which is consistent with the optical observation (inset).

Figure 6c, d compares the SOC distribution when charged at 0.05C and 1C for the 50 μm (2 mAh cm$^{-2}$) and 100 μm-thick (4 mAh cm$^{-2}$) electrodes respectively (Top: separator; Bottom: current collector). At low C-rate, the particle phase separates between Stage 2 (SOC = 0.52)/Stage 3 (SOC = 0.23) at the global SOC = 40%, and Stage 2/Stage 1 (SOC = 0.95) at the global SOC = 70% both at the electrode (inter-particles) and particle (intra-particle) level, with clear intra-particle phase boundaries. It is noted that even at such a low C-rate, large particles in the vicinity of the separator in the thick electrode still display a shrinking core lithiation process (indicated by the black arrow), as a consequence of the inhomogeneous current density due

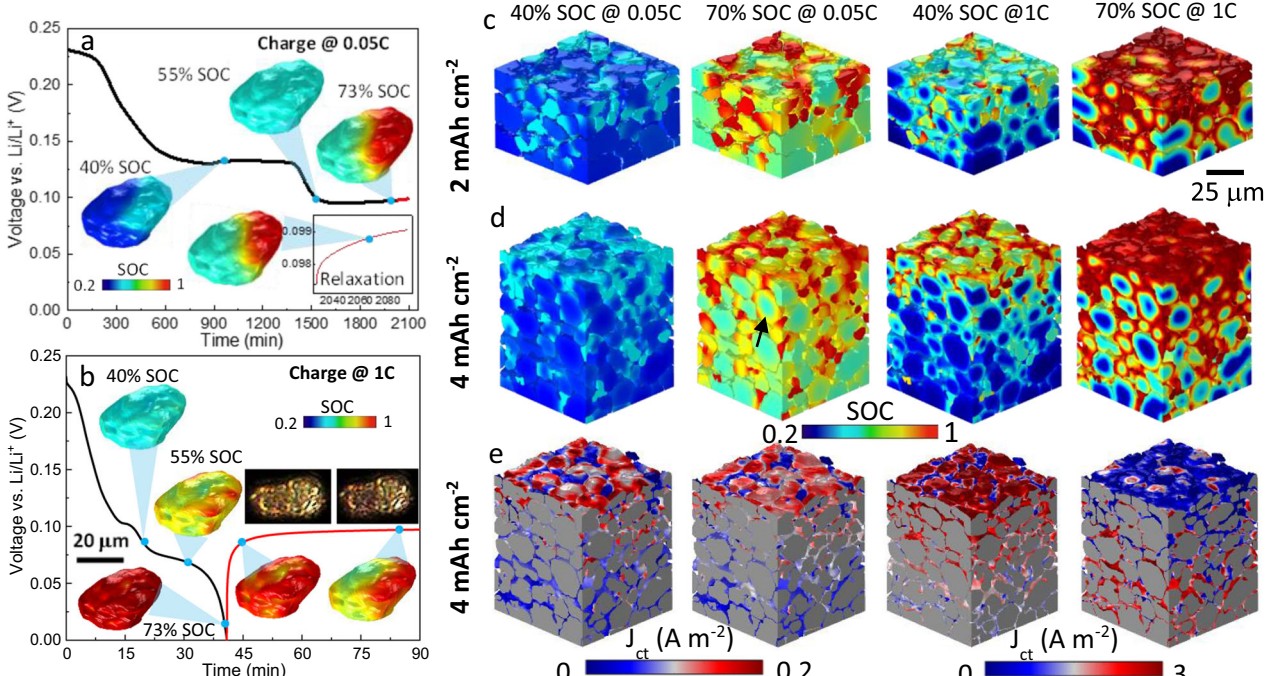

**Fig. 6 | Comparison of lithiation and phase separation dynamics in a single particle and electrodes at different C-rates. a** Simulated voltage curve shown with phase separation governed by intercalation wave mechanism at 0.05C charge, followed by relaxation in a single particle; **b** simulated voltage curve shown with shrinking core behavior of the single particle at 1C charge followed by lithium redistribution during relaxation; **c**, **d** spatial distribution of SOC when charged to different global SOCs at different C-rates for 2 mAh cm⁻² (50 μm) and 4 mAh cm⁻² (100 μm) electrode respectively; the black arrow in (**d**) points out the particle that exhibits shrinking core behavior; **e** charge transfer current density distribution in the 4 mAh cm⁻² electrode (note the different ranges of current density used in the colourmap at 0.05C and 1C). Source data are provided as a Source Data file.

to the electrolyte concentration gradient along the thickness direction. This becomes more predominant at 1C, where the particles at the electrode/separator interface (top) show significant shrinking core behavior with a lithiation gradient both inside the particles and along the thickness direction in the 4 mAh cm⁻² electrode.

Figure 6e compares the reaction kinetics at different charge rates in the thick electrode. Charging at 0.05C, a preferential reaction is observed to take place at the corner and sharp edge of the particle (a clear definition of particle corner and edge is shown in Fig. S7), consistent with the observation from the optical experiment (Fig. 3f, Particle C-1, C-2). As the charging current increases to 1C, a pronounced reaction gradient exists along the thickness direction: particles in the vicinity of the separator (top) are lithiated much faster than the bottom ones owing to the gradient of electrolyte concentration (Fig. S10) due to the large tortuosity of Li⁺ transport in the thickness direction (Fig. 4c, d) caused by the low porosity ($\varepsilon = 0.28$, Fig. 4e) and horizontally aligned particles (Fig. 4f); however, the reaction front propagation is observed at the late stage of charging (70% SOC), which could be described as an "electro-autocatalytic" behavior. This reaction front propagation phenomenon was also reported in the NMC electrode[46]. It is widely believed that lithium plating in thick graphite electrode is dominated by ion transport in electrolyte[47] and therefore the tuning of porosity and tortuosity is prioritized; however, as it is observed here, the majority of lithiation current/flux resides at the electrode/separator interface due to the electrolyte concentration gradient, implying that the optimization of graphite particle morphology is equally important since unfavorable geometries will cause surface crowding, and thus large overpotential at the separator/electrode interface and trigger early and heterogeneous lithium plating. This will be further discussed in the following section.

Figure 7 presents the first known results correlating lithium plating and its thickness with the phase separation and 3D architecture of the electrode. Figure 7a shows the predicted plated area where the

plating reaction is kinetically possible during charging at 1C. This is consistent with the previous research that reported lithium plating in the graphite anode with a modest mass loading (2.2 mAh cm⁻²) at the maximum charging rate of 1C in the commercial pouch cell format[44,48], whereas the onset of early plating and the relationship of the micro-dynamics with the microstructure was not well understood. Here, the predicted early onset of plating is found at 65% SOC (Fig. 7a), agreeing well with the experiment detected by the OCV relaxation method shown in Fig. 5c. This is also substantiated by operando synchrotron X-ray radiography in a 2 mAh cm⁻² graphite electrode, where plating layer is observed at the electrode/separator interface at 60% SOC under 1C charging (Fig. S9). Complementing the experiment, Fig. 7a reveals that plating nucleates at the edges and sharp corners of the particles. As the charge continues, not only does the plated region grow at the particle surface but also into the deeper area of the electrode. The spatial visualization of the plating thickness in Fig. 7b provides a more practical representation of the heterogeneous lithium plating. The plating thickness increases from 6 nm at 65% SOC to 20 nm at 75% SOC, preferentially on the surface of small particles or at the edges/sharp corners of large particles. The former is attributed to an early saturation amongst bigger particles, and the latter is due to the sluggish solid-state diffusion, particularly at the edges and corners where the surface area to volume ratio is the largest and thus experi-ence a higher inward Li⁺ flux. With the increase of the electrode thickness (4 mAh cm⁻², Fig. 7c), electrolyte concentration gradient builds up causing a gradient of lithiation current density (as shown in Fig. 6e), so preferential plating becomes more remarkable. Hetero-geneous electrolyte distribution is observed between neighboring particles arising from the uneven distribution of the pore size (Fig. 4b); however, this does not show a direct link with the plating propensity compared to surface saturation (Fig. S10). Global and local SOC ana-lysis proves that plating occurs when the maximum SOC reaches -0.98 (indicated by the inflection of the solid curves in Fig. S11a). This is

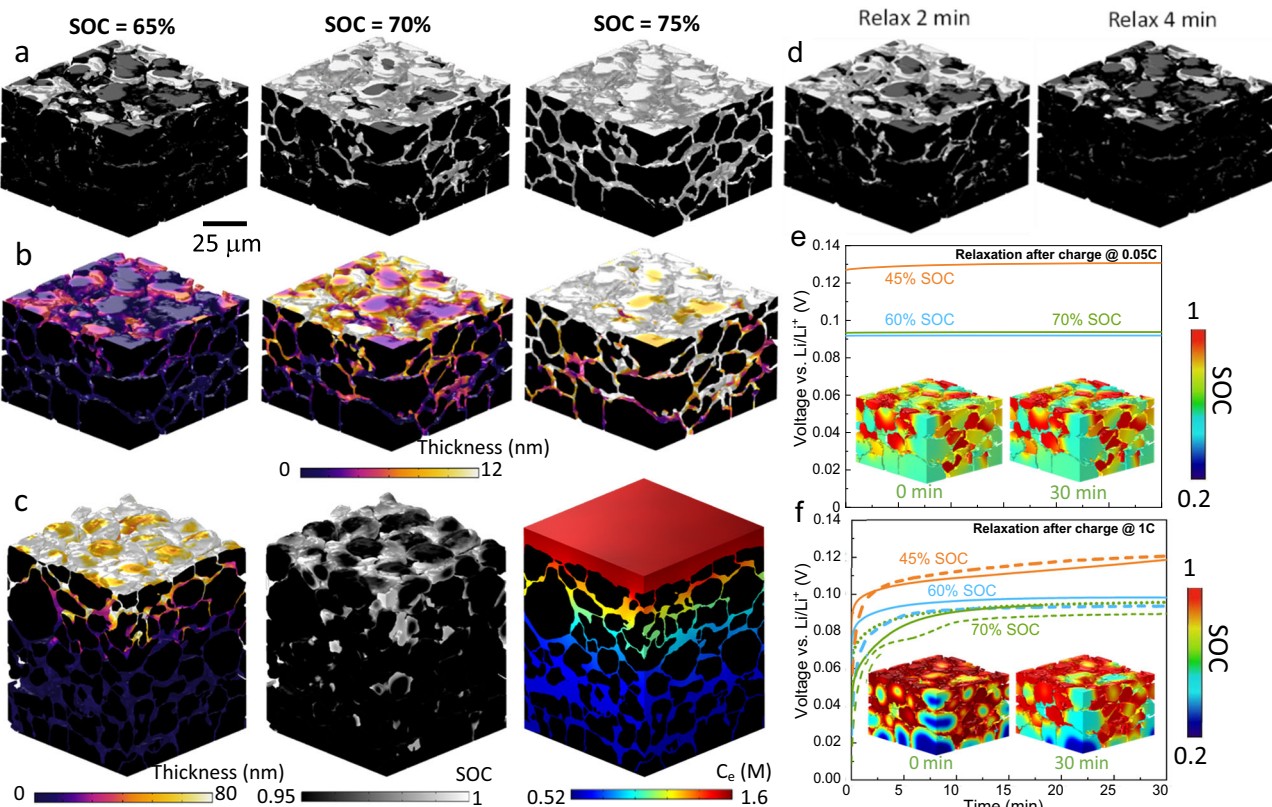

**Fig. 7 | The onset of lithium plating and its spatial distribution predicted by 3D microstructure-resolved phase field model. a** Spatial distribution of lithium plating (white) and (**b**) the corresponding plating thickness predicted in the 2 mAh cm⁻² (50 μm) electrode charged at 1C; in comparison, the plating thickness in a 4 mAh cm⁻² (100 μm) electrode is shown in (**c**), alongside the SOC and electrolyte concentration $C_e$ distribution at 65% SOC; **d** stripping vs. time after 1C

charge to 70% SOC; **e**, **f** voltage and SOC relaxation after 0.05C and 1C charge respectively (solid: simulation; dashed: experiment; green dotted: simulation with stripping deactivated), with the insets showing the SOC evolution over 30 min relaxation after charging to 70% SOC. Source data are provided as a Source Data file.

substantiated by operando optical imaging of the graphite particles at the LSE, where the plating was observed to nucleate first in the particles of highest local SOC[37]. The lithiation current decreases as the plating onset. Charging the cell at 1C to 100% SOC will cause 17% capacity loss (Fig. S11a), which drops to 1.3% and 0.2% if a voltage cut-off is set at 75% and 65% SOC respectively, corresponding to a plating thickness of 20 and 3 nm. However, this is not an issue since, practically, the CC regime only lasts for 36 min (60% SOC, Fig. S11b) before hitting the cut-off voltage (5 mV vs. Li) and merely avoids the onset of plating.

**Resting for being faster: advanced charge protocol informed by relaxation dynamics**
Beyond the phase separation and plating risk during fast charge, the 3D microstructure-resolved model can reveal the relaxation dynamics both at the electrode and intra-particle level to provide physics-based evidence of a rational OCV step to be embedded into advanced fast charging protocols such as adaptive pulse charge[32,34] and multistage constant current charge[35,49,50] (or similar) used by most of the battery and EV manufacturers. Current design strategies of fast charge protocols are either guided by experiment[33,51,52] or machine-learning algorithms[53,54], lacking a physics-based understanding of the electrode dynamics, which can be assisted by the 3D phase-field modeling presented here.

Figure 7d shows the simulated stripping process during relaxation after being charged to 70% SOC. The plating thickness drastically decreases within the first 4 min. Figure 7e, f compares the voltage profiles (solid: simulation; dashed: experiment) and relaxation

behavior after being charged to different SOCs at 0.05C and 1C respectively. Insignificant change is observed for the former as the charging at low C-rate is under equilibrium; however, a substantial response both in the lithium redistribution and voltage recovery is noted after 1C charge. Consistent with Fig. 6b, the lithiated large graphite particles transform from a core-shell pattern to a phase-separated distribution, accompanied by minimization of the phase boundaries, saturated surface and under-lithiated core, entering a more energetically stable state. The relaxation process at the early stage (70% SOC) is less drastic, arising from the stripping (time length coincides with Fig. 7a) and lithium redistribution. To highlight the effect of stripping in suppressing equilibration of the electrode, the relaxation profile predicted with lithium stripping deactivated (green dotted line) is also added for comparison.

The results in Fig. 7d–f not only suggest how long the relaxation time should be to complete the stripping of any plated lithium that is active (i.e., in contact with the electrode matrix), but also more crucially, when a rest step should be placed in the fast charge protocol to proactively reduce the plating risk, meanwhile effectively shortening the total charging time (reducing the CV regime). Figure 8a plots the graphite OCV as a function of the stoichiometry, with the symbols indicating a variety of fast charge timesteps followed by a rest step. The insets display the SOC distribution before/after a 3-min relaxation and the corresponding SOC_max as a function of relaxation time is shown in Fig. 8b. At 19% SOC, the surface concentration of lithium equilibrates almost immediately due to the low concentration gradient and relatively fast solid-state diffusion at low SOC (Fig. S12). At 28% global SOC, SOC_max at the particle surface reaches 0.4 (i.e., Stage 2 has

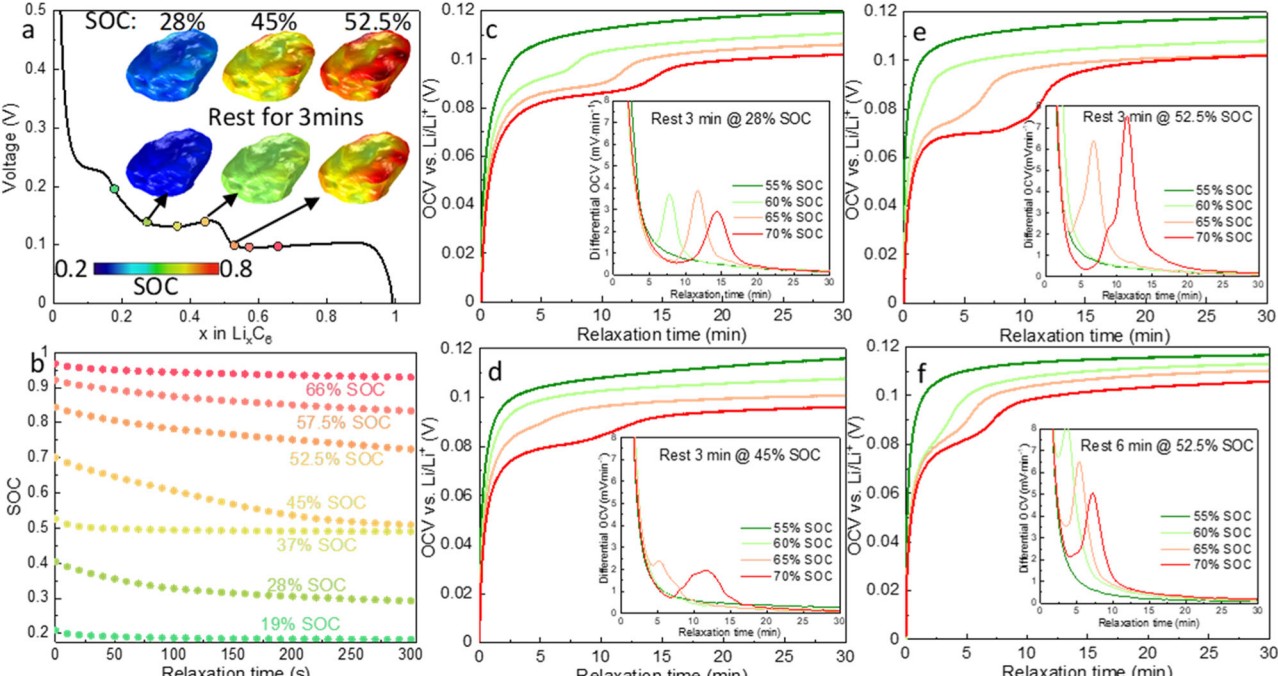

**Fig. 8 | Physics-based design and experimental validation of a rest step implemented in fast charge protocols. a** Predicted voltage curve shown alongside the SOC distribution in a single particle before and after a 3-min rest after being charged to different SOCs at 1C; **b** predicted SOC$_{max}$ equilibration of a single particle as a function of relaxation time; **c**–**e** experimentally measured OCV relaxation of the 2 mAh cm$^{-2}$ coin cell after being charged to different SOCs at 1C, following a 3-min rest step at 28%, 45% and 52.5% SOC respectively. The insets display the differential OCV; **f** experimentally measured OCV relaxation after being charged to different SOCs following a rest step at 52.5% SOC for 6 min. Source data are provided as a Source Data file.

not nucleated), which is unstable and diminishes rapidly (also see the inset in Fig. 8a), gradually relaxing toward Stage 3. As charge continues to 37% global SOC, SOC$_{max}$ reaches 0.5 and exhibits an instant equilibration. A rest step after charging to 45% global SOC shows a similar but more drastic relaxation behavior compared to 28% SOC, with the SOC$_{max}$ dropping from 0.7 (unstable) to 0.5 (Stage 2) within 200 s (inset in Fig. 8a). At 52.5% global SOC, the particle surface is highly unstable, consistent with the OCV relaxation curve shown in Fig. 5c. The redistribution of SOC does not finish within 3 min (inset in Fig. 8a). Stage 1 is only observed to nucleate when the global SOC reaches 60%, and thereafter the equilibration process is significantly facilitated because the unstable intermediate SOCs can phase separate into Stage 1 and 2 from now on. However, this is not conducive to battery safety and fast charge. From the perspective of delaying surface saturation, and thereby early onset of plating, a rest step at 28, 45 or 52.5% SOC seems to be most beneficial.

To validate this, plating behavior is compared by monitoring the OCV relaxation of the coin cells after being charged to different SOCs following the rest step at 28%, 45% and 52.5% SOC respectively. Figure 8c displays that a 3-min rest at 28% SOC does not delay the onset of plating compared with the previous test (see Fig. 5c). A break at 28% SOC also benefits little to the electrolyte concentration gradient since it is not fully developed in the early stage. In contrast, it is delayed until 70% SOC when a rest step is implemented at 45% SOC for 3 min (Fig. 8d). Resting at 52.5% SOC does not delay plating (Fig. 8e) either, even after doubling the rest time (Fig. 8f). In addition, note that the 70% SOC curve in Fig. 8e shows a lower and flatter voltage plateau than those in Fig. 8c, d, which is speculated to be due to a larger stripping current as a consequence of more extensive plating. As an indicative example of this phenomenon, charging to 52.5% SOC followed by a rest step cannot effectively reduce the extent of surface saturation. Another explanation is that Stage 1 may have already nucleated at 52.5% SOC in some of the particles, considering the SOC gradient along

the thickness direction, weakening the effect of surface equilibration. The insets of Fig. 8c–f show the differential OCV vs. time, wherein the peak position indicates the stripping period, and the peak height represents the stripping current. It is observed that after resting at 28% the longest stripping period is required, while the 52.5% SOC case shows the largest stripping current. Extending the rest period from 3 to 6 min can reduce both metrics (Fig. 8f), but neither is effective in delaying the onset of plating.

To summarize, this section provides a proof-of-concept study illustrating how an efficient relaxation step during fast charging can (1) mitigate the lithium concentration gradient in the graphite particles and alleviate plating; (2) facilitate capacity recovery by a thorough stripping of the plated lithium that is active and reversible; (3) reduce capacity loss arising from the formation of dead lithium; (4) shorten the CV regime where the concentration relaxation is significantly less efficient. The combined experiment and modeling results suggest that a 3-min rest at 45% SOC is appropriate to a charging rate up to 3C for the electrode with an areal capacity up to 2 mAh cm$^{-2}$, and 1C for a 3 mAh cm$^{-2}$ electrode, as the resistance from ionic transport in the electrolyte is insignificant. However, the optimal relaxation time shifts to 30% SOC and 20% SOC charging at 2C and 3C respectively for the 3 mAh cm$^{-2}$ electrode (Fig. S13), owing to an increased lithiation current at the electrode/electrolyte interface with the development of concentration gradient of the electrolyte. This new perspective building on the physics-based evidence obtained by our 3D microstructure-resolved phase-field model is anticipated to provide new insights into the development of advanced fast charge protocols such as the multi-step constant-current (MSCC) profiles[49,50] used by most of the EV OEMs. Specifically, there is a great potential to enforce a prolonged preliminary charging to higher SOC for a more efficient OCV step (relaxation of concentration gradient in both the particles and electrolyte), conduct a rational refinement of the ratio between each CC steps, and

even develop a new pattern of MSCC with modulated CC-OCV steps to further reduce the charging time.

## Discussion

This study aims to reveal the spatial dynamics of lithiation and plating/stripping in phase-separating graphite electrodes via coupled operando optical microscopy, 2D macroscopic and 3D microstructure-resolved phase-field graphite modeling techniques, to assist the optimization of electrode microstructure and rational design of advanced fast charge protocols to facilitate the fast charge performance of graphite-based anodes for next-generation automotive batteries. It is found that, from the separator to the current collector, graphite particles spatially follow the order: saturated Stage 1 (metastable)−Stage 1/2 (stable)−phase boundary (unstable)−Stage 2/3 (stable)−phase boundary (unstable)−Stage 3 during fast charge. The stage distribution and phase boundaries remain static during relaxation after low current charging; in contrast, the phase boundaries continue to propagate along the thickness direction during relaxation after high current charging due to inter-particle lithium exchange and lithium stripping. The occurrence of plating reduces the effective intercalation current. The particle orientation and shape dictate the intra-particle lithiation directionality at low C-rates, which is dominated by an intercalation wave mechanism. In contrast, a non-equilibrium shrinking core mechanism suppressing the phase-separating behavior is predominant in large particles or under high C-rates charging due to the rate-limiting solid-state mass transport. This leads to intra-particle heterogeneous SOC distribution and coexistence of Stage 1/2/3, which is highly likely to trigger localized early plating.

An energy barrier of 20 mV for plating nucleation is determined based on both experiment and modeling results. Plating preferentially nucleates at the corners and edges of the graphite particle, irrespective of the particle orientation. A surface saturation of lithium content (-SOC = 0.98) is the prerequisite condition for the onset of plating, regardless of the charging C-rates. Once the plating energy barrier is overcome, the majority of charge transfer current contributes to the plating reaction instead of intercalation into the graphite. Fully charge the electrode at 1C will cause 17% capacity loss (albeit largely reversible loss), and this value drops to 1.3% and 0.2% if a voltage cut-off is set at 75% and 65% SOC respectively, corresponding to a plating thickness of 20 and 3 nm.

A distinct difference in relaxation dynamics is found under low and high-rate charging. The SOC distribution and phase boundary remain static during relaxation after low C-rate charging. In contrast, a drastic SOC redistribution occurs at relaxation after 1C or higher C-rates charging. As a result, at the electrode level, unstable lithium de-intercalates and then intercalates at the phase boundary, which keeps propagating during relaxation; at the particle level, the SOC redistributes from a core-shell structure to a phase separated arrangement to minimize the phase boundary. However, if Stage 1 nucleates at the particle surface, the inter-particle lithium exchange will be suppressed, and the unstable phase eventually phase separates within the particle, which is not effective in ameliorating surface saturation and delaying lithium plating. Assisted by the combination of experiment and 3D microstructure-resolved phase-field modeling, we conclude that it is beneficial to embed a rest period for 3 min at 45% SOC for a charging rate up to 3C for the electrode with an areal capacity up to 2 mAh cm$^{-2}$. For a 3 mAh cm$^{-2}$ electrode, the optimal relaxation time is at 45% SOC, 30% SOC and 20% SOC under 1C, 2C and 3C charging respectively. The efficient relaxation step helps to (1) efficiently relax the lithium concentration gradient in the graphite particles; (2) facilitate capacity recovery by a thorough stripping of the plated lithium that is active and reversible; (3) reduce capacity loss arising from the formation of dead lithium; (4) shorten the CV regime where the concentration relaxation is significantly less efficient.

## Methods

### Materials

Natural spherical graphite (Superior Graphite, USA) particles (95.5 wt %) were mixed with CMC binder (1 wt%), SBR binder (1.5 wt%) and conductive carbon C65 (2 wt%) to produce the electrode sheets with the capacity loading from 1 to 4 mAh cm$^{-2}$, corresponding to the electrode thickness of 25, 50, 75 and 100 μm respectively. The mass loadings are 3.5, 7, 10.5 and 14 mg cm$^{-2}$ respectively. The copper current collector is 18 μm thick.

### Operando optical microscopy

The electrode of the areal capacity 2 mAh cm$^{-2}$ was cut into a 2.2 × 8 mm$^2$ rectangle and assembled into the ECC-Opto-Std optical cell (EL-cell, Germany). The spatial arrangement of the optical cell, including the glass window, graphite electrode, 250 μm-thick glass fiber separator, lithium foil and sealing O ring is schematically shown in Fig. 1. The cell was then placed under a high-resolution Keyence VHX-7000 optical microscope (UK) using the objective lens of 700x. A potentiostat (Gamry Instruments, USA) was used to cycle the optical cell at 2 and 4 mA cm$^{-2}$ (lateral current density), synchronized with the camera that record the dynamic process of lithiation, rest and delithiation cycle at the frequency of 1 min/frame.

### Electrochemical cycling for OCV relaxation measurement

Anodes of the capacity 1, 2 and 3 mAh cm$^{-2}$ were cut into 15 mm diameter and assembled into coin cells with Celgard 2325 separator (diameter: 19 mm) and Li metal (diameter: 15.6 mm, thickness: 250 μm). In total, 100 μl of 1 M LiPF$_6$ in 30 vol. % ethylene carbonate (EC) and 70 vol. % ethyl methyl carbonate (EMC) + 2 wt% vinylene carbonate (VC, Soulbrain, USA) were used as the electrolyte. Two cycles of formation process were performed on BCS battery cyclers (Biologic, France): starting with a C/20 charge (constant current, CC) followed by a constant voltage (CV) step at 5 mV (C/40 cut-off). The cell was then discharged to 1.5 V at C/20. After the formation step, the cell was fast charged to a variety of SOCs at constant step size (30%, 35%, 40% ....) with each fast charge step followed by a 30-min rest to record the OCV relaxation and then a discharge step to 1.5 V. The test was performed on electrodes of different thicknesses at 1C, 2C and 3C respectively at 25.8 °C.

### X-ray computed tomography and 3D image analysis

The graphite electrodes of different thicknesses (50 and 100 μm, corresponding to 2 and 4 mAh cm$^{-2}$) were cut into disks of ~500 μm in diameter using a laser cutter (A Series/Compact Laser Micromachining System, Oxford Lasers, Oxford, UK). The electrode disks were then scanned at 0.2 μm voxel size using a lab-based X-ray micro-CT system (Zeiss Xradia Versa 520 X-ray microscope, Carl Zeiss, CA, USA)[55] at the X-ray tube voltage of 80 kV (tungsten emission). In total, 2001 projections were collected over the 360° rotation of the sample. The projections were then reconstructed using commercial software (Zeiss XMReconstructor) employing a proprietary Feldkamp-Davis-Kress (FDK) algorithm[56]. Microstructural characterization of the electrode was carried out in commercial software Avizo V9.5 (Avizo, Thermo Fisher Scientific, Waltham, Massachusetts, U.S.). The geometrical metrics of graphite particles (orientation, size and shape etc) were analyzed by extracting the moments of inertia of its ellipsoid template and the eigenvalues of the covariance matrix[57].

### 1D + 1D homogenized phase-field model

The 1D + 1D phase-field model in this study derived from the mathematical model developed by the Bazant group[58–60], wherein partial differential governing equations (PDEs) and conservation of species (i.e., Li$^+$, PF$_6^-$, e$^-$ and Li) are assigned to different phases that are spatially lumped together in the entire simulation domain. The potential in the electron-conducting phases (graphite + conductive carbon, $\mu_e$) is

solved by:

$$\nabla \cdot J_e = 0 \tag{1}$$

$$J_e = -\sigma_e \nabla \mu_e \tag{2}$$

where $\sigma_e$ is the electrical conductivity and $J_e$ represents the electron flux. The lithium concentration in graphite particles ($c_s$) is solved according to a microscopic mass balance along the particle radius as:

$$F\frac{\partial c_s}{\partial t} + \nabla \cdot J_s = 0 \tag{3}$$

where $F$ is the Faraday constant. Instead of using concentration gradient ($\nabla c_s$), the mass transport in the graphite in a phase-field model is governed by the gradient of chemical potential ($\widetilde{\mu}$), which is a derivative of free energy $G$ (i.e., as a function $c_s$ and $\nabla^2 c_s$) to capture the uphill diffusion and phase separation (detailed mathematical expression can be found in Supplementary Method 1). The Li flux in the graphite particle $J_s$ is expressed as:

$$J_s = -FD_s/RT \cdot (1 - \widetilde{c}_s) c_s \nabla \widetilde{\mu} \tag{4}$$

where $D_s$ represents the solid-state diffusion coefficient, $R$ is the gas constant, $\widetilde{c}_s$ is the normalized lithium concentration in the graphite ($\widetilde{c}_s = c_s/c_s^{max}$). The potential and electrolyte salt concentration in the pore phase ($\mu_p$ and $c_e$) are solved according to the concentrated solution theory[61,62] as follows:

$$\varepsilon^* F\frac{\partial c_e}{\partial t} + \nabla \cdot J_p = 0 \tag{5}$$

$$-\varepsilon^* F\frac{\partial c_e}{\partial t} + \nabla \cdot J_n = 0 \tag{6}$$

where $\varepsilon^*$ is the porosity. $J_p$ and $J_n$ stand for the Li$^+$ and PF$_6^-$ flux, respectively, and are expressed as:

$$J_p = -t_p \frac{\varepsilon^*}{\tau}\sigma_{io} \nabla \mu_p \tag{7}$$

$$J_n = \frac{1}{t_p}F\frac{\varepsilon^*}{\tau}D_{io}\nabla c_e - \left(1 - t_p\right)\frac{\varepsilon^*}{\tau}\sigma_{io}\nabla\mu_p \tag{8}$$

where $t_p$ is the transference number, $\tau$ is the tortuosity factor, $\sigma_{io}$ and $D_{io}$ stands for ionic conductivity and diffusion coefficient respectively. Butler-Volmer kinetics accounting for the SEI kinetics and resistance[8] are used to describe the charge transfer at the electrode/electrolyte interface. The charge transfer current density $J_{ct}$ is expressed as:

$$J_{ct} = i_0\left[\exp\left(\frac{\alpha F}{RT}\left(\mu_e - \mu_{psurf} - V_{eq}\right)\right) - \exp\left(-\frac{(1-\alpha)F}{RT}\left(\mu_e - \mu_{psurf} - V_{eq}\right)\right)\right] \tag{9}$$

where $i_0$, $\mu_{p\_surf}$ and $V_{eq}$ represent the exchange current density, Li$^+$ potential at the graphite surface and equilibrium potential (open circuit potential, OCV) respectively, and their detailed mathematical expressions can be found in Supplementary Method 1. $V_{eq}$ is derived from a simplified 1D graphite free energy model as described by Bazant's group[58,60], rather than from the two-layer model[58], or various generalizations to 3 or more periodic layers, which are more accurate in modeling the phase diagram at different temperatures[63,64]. The model is implemented in COMSOL Multiphysics V5.6. A detailed mathematical description of the model, parameters and boundary conditions are presented in the Supplementary Method 1.

### 3D microstructure-resolved phase-field model
The 3D datasets of the graphite electrodes obtained from X-ray CT scans were firstly segmented into graphite and pore/CBD phases using marker-based watershed segmentation[65], followed by adaptive meshing using the commercial software package Simpleware ScanIP N-2018.03. The physical size of the mesh element was adjusted according to the feature size of each phase and conformal nodes were ensured at the boundaries. The 3D phase-field model is easily adapted from the 1D + 1D model and no volume-averaged material parameter is used. A detailed mathematical description of the model, parameters and boundary conditions are presented in the Supplementary Method 1.

## Data availability
The data generated in this study are available in the main text and Source Data file. Source data are provided with this paper.

## Code availability
The code used in the study is available from the corresponding author upon request.

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

## Acknowledgements

This work was supported by the Engineering and Physical Sciences Research Council [EP/X000702/1, EP/R020973/1, EP/M028100/1]; and the Faraday Institution (faraday.ac.uk; EP/S003053/1, grant numbers FIRG014, FIRG024, FIRG025 and FIRG028). P.R.S. acknowledges funding from the Royal Academy of Engineering (CiET1718\59). X.L. thanks the advice provided by Dr. Huada Lian from MIT.

## Author contributions

X.L. and P.R.S. conceived the study. X.L., A.B., and M.L. conducted modeling; S.D. and M.Z.B. assisted with the model development; R.E.O. and X.L. conducted operando microscopy experiment; Q.L. assisted with experiment design and analysis of OCV relaxation; K.O.R. and E.K. performed the rate capability test and OCV measurement of the graphite electrodes; X.L. and A.W. performed X-ray CT scans of the electrode; D.P.F. advised on the data analysis and result interpretation; D.J.L.B. guided model validation and data analysis; X.L. drafted the manuscript and all co-authors reviewed the manuscript.

## Competing interests

The authors declare no competing interests.

## Additional information

¹Electrochemical Innovation Lab, Department of Chemical Engineering, UCL, London WC1E 7JE, UK. ²The Faraday Institution, Quad One, Harwell Science and Innovation Campus, Didcot OX11 0RA, UK. ³School of Engineering and Materials Science, Queen Mary University of London, London, UK. ⁴Department of Civil and Industrial Engineering, University of Pisa, 56122 Pisa, Italy. ⁵Department of Chemical Engineering, MIT, Cambridge, MA 02139, USA. ⁶MOE Key Laboratory of Enhanced Heat Transfer and Energy Conservation, Beijing University of Technology, Beijing, China. ⁷School of Metallurgy and Materials, University of Birmingham, Birmingham B15 2TT, UK. ⁸National Renewable Energy Laboratory, 15013 Denver West Parkway, Golden, CO 80401, USA. ⁹Department of Mathematics, MIT, Cambridge, MA 02139, USA. ¹⁰Department of Engineering Science, University of Oxford, Parks Road, Oxford OX1 3PJ, UK. ✉e-mail: xuekun.lu@qmul.ac.uk; Paul.shearing@eng.ox.ac.uk

