## [Peer Review File · Nature Communications]

nature portfolio

Peer Review FilePeer review comments, first round review

Reviewer #1 (Remarks to the Author):

This article describes a multi-modal study of graphite anodes during charging and OCV rest periods. Gradients in SOC are observed and modeled at both the intra-particle and inter-particle levels. The influence of particle size, shape, and orientation, are discussed on local heterogeneity. Finally, the insertion of 3-min OCV rest periods are studied to allow for SoC relaxation to delay Li plating.

Overall, this is an interesting article that will provide value to the fast-charging community. In particular, the discussion of intra-particle SOC gradients is under-discussed in the literature, and provides high impact. Observations of individual particle shape and orientation effects are interesting and relevant to understand heterogeneity at small length scales. However, there were several confusing and unclear aspects to the discussion, which made the paper hard to follow at times. The reviewer is supportive of publication in Nature Communications, after the following points are addressed:

1) The definition of C-rate and current density during the operando optical microscopy experiments were unclear and difficult to follow at several points in the discussion. First: what area was used to define current density in these experiments? As discussed throughout, the local current density actually varies spatially throughout the electrode surface in thick electrodes. Typically, an areal current density is defined, when the area of the working and counter electrodes are the same, and are placed in a parallel electrode geometry. However, in this optical experiment, it was not clear what direction the electric field is pointing (is it normal to the electrode thickness, or in the lateral direction?) Figure S1 was difficult to interpret, with respect to the location of the working and counter electrodes, and the perspective from which the optical imaging is taken. A more detailed schematic with labels and arrows showing these directions would be helpful. Based on these difficulties in visualizing the setup, the discussion of "thickness" and "interface" were also confusing, as discussed in a later comment. A significant part of this challenge is that the actual optical cell geometry, with the working, counter electrodes, separator etc. are in the SI. The reviewer feels that this information should be moved to the main text, and the definition of current density and C-rate need to be more clearly explained for the optical cell. For example, the authors state "Here a current density of 2 mA cm⁻² corresponds to the current density of 1C in the coin cell experiment of this study, but the effective C-rate of the former is much lower due to the large geometry of the electrode". What is meant by "effective C-rate"? Please define current density and C-rate more mathematically for the optical cell, with visualizations of the electric field and electrode orientations in the main text.

2) Following on point 1, how is C-rate defined in the optical experiments? Typically, the C-rate is based on the charge current needed to fully charge the electrode, based on the areal capacity (e.g. 2 mAh/cm²) and corresponding current density. However, it was not clear what the areal capacity is in Fig. 1. The caption lists "the first 400 microns of a total thickness of 2 mm". Was the thickness of the electrode really 2mm? Or is this the "lateral thickness"? If the later, how is C-rate defined? This was confusing.

3) The paper discusses "global SOC" several times, but this term is not clearly defined. Is this referring to the charge of the electrode relative to the theoretical capacity of the entire electrode? Please explain how this is used in the various configurations (optical vs. coin cell, etc.).

4) Why was the plated Li in Fig. 1 so out of focus? And in which direction (relative to the microscope and electrode configuration) is the Li plating occurring? Is it constrained to only grow towards the separator (as would be typical in a coin cell), or is it growing out towards the microscope objective? Please clarify this in the text.

5) The text refers to the EEI as the "electrode electrolyte interface". Does this mean electrode-separator interface? This terminology is confusing, because there is electrolyte present throughout the porous electrode geometry. Also, is there any electrolyte on the top surface (towards the microscope objective) that is not within the pores of the electrode? If so, wouldn't there be concentration gradients and electrolyte transport effects within this "top" region (perpendicular to

the electrode-separator interface)? This was not clear, and again, might be more clearly shown when adding the cell geometry information to the main text.

6) In the paragraph below figure 1, "nominal current density" is used for the first time. What is meant by this, and how does it relate to "effective C-rate" described above?

7) Overall, the reviewer found the optical microscopy section the most difficult to follow- in the paper. It was not clear how some of these terms were defined (C-rate, current density, etc." relative to the areal capacity fo the porous electrode relative to the current collector, vs. the direction of lithiation. It should be emphasized to the reader if this visualization is actually in the in-plane (lateral direction) of a "typical" electrode, and that the corresponding direction of SoC gradients observed is perpendicular to what is normally considered the "thickness" direction, which was unclear. That would help to clarify several of the points above.

8) All of the experiments in this paper appear to be 2-electrode measurements with a Li metal counter electrode and no reference electrode. Therefore, in the experimental voltage curves, although the graphite potential is plotted as "vs Li/Li+", the effects of overpotentials at the Li metal counter electrode (which is not at 0V during non-zero currents) and IR drop will impact the measured potential. It was not clear, therefore, how it could be determined that 20 mV is the energy barrier for nucleation. This was also confusing, because it was not clear if the authors were stating that the electric potential of the composite graphite anode (composed of many particles at different SOC) is the nucleation barrier, or if this barrier applies to the electrochemical potential of individual graphite particles, which are at different SOC. Please clarify how this single number is used in this definition, and how it relates to inter-particle SoC heterogeneity.

9) The discussion of the individual particle geometries, orientations, size effects etc. was great, and a true highlight of the paper. One thing that was not clear is the term "Li flow direction". What is meant by that? Are the authors referring to the vector pointing normal to the electrode/separator interface? Li is clearly "flowing" in 3-dimensions within this 3-d electrode geometry, so this terminology was not clear. How does this "flow direction" relate to the electric field direction in the electrolyte phase surrounding these individual particles? It seems that the electrolyte potential gradients will also lead to a 3-D distribution of the electric field in the vicinity of the individual particles, so it was not clear what the "flow direction" indicates.

10) The authors state that "intra-particle SOC heterogeneity cannot be captured by macroscopic modelling techniques and yet is the major cause of the early onset of lithium plating". What is meant by "the major cause"? Inter particle heterogeneity (including SoC gradients throughout the electrode thickness) lead to local current focusing, which accelerates intra-particle heterogeneity. So it seems that both intra- and inter-particle SoC heterogeneity are coupled in a composite electrode, and we can not state that one is the "major cause" vs. the other.

11) The 3-D microstructure characterization section provides details that do not seem to directly relate to the rest of the story of the paper. For example, it was not clear how the details in Fig. 3d-j directly relate to the discussion? Perhaps the most relevant point is the significant differences in tortuosity factor in the vertical vs. horizontal directions. It is worth mentioning that the optical cell geometry will thus experience greater electrolyte transport limitations than the coin cell, if the direction of "thickness" is in the horizontal direction.

12) In discussing the OCV profile, the authors state that "The inflection of the OCV curve is known to be an indication of lithium plating^{42, 43}. The curve is divided into two plateaus, the first of which corresponds to the stripping process and the dissolved lithium back-intercalating in a deeper area of the electrode to maintain neutrality, thus slowing down the rise of the OCV curve and delaying the equilibration.". This plateauing behavior during OCV, and the deflections associated with stripping of plated Li, re-intercalation, and the delayed onset of equilibration are described in detail in Ref. 37, which would be good to cite here. In general, Ref. 37 provides a few relevant observations that further strengthen the discussion in the current paper, including a direct observation that Li plating nucleates on individual particles that lithiate fastest (the particles that experience the highest local SOC and turn gold first are the same particles where Li plating nucleates"), as well as the fact that the OCV peaks shift to longer times when the C-rate and thickness increase (discussion of Fig. 4).

13) Fig. 4a was difficult to interpret, as the panel is quite small with several labels and multiple voltage curves. It would be helpful to increase the width of this panel in the figure, to improve readability.

14) It was a bit surprising to see significant Li plating at a 1C rate for relatively "modest" electrode loadings of 1-2 mAh/cm². Do typical commercial Li ion batteries experience plating at 1C for these loadings? It would be useful to benchmark these observations of plating at 1-2C rates for 1-2

mAh/cm² loadings with other reports in the literature; for example K. G. Gallagher et al., J. Electrochem. Soc. 2016, 163 (2), A138–A149.

15) No SOC “color bar” was provided in Fig 5a. Is it the same color scale as Fig. 5b? In general, the use of a color scale to represent SoC and current density in the same figure (Fig. 5) makes the figure a bit confusing to follow, especially with two different color scale bars for current density in panels c-e.

16) In the later parts of the paper, a CC-CV charging protocol with a cutoff voltage in graphite potential was introduced, but this was not clearly explained: was this the first time that a CC-CV protocol was used? And was it only used on some experiments in the paper?

17) In the summary of the charge-protocol section, it is stated that the section provides proof that a relaxation step can allow for “complete stripping of any heterogeneous early plating”. It was unclear how this was proven? Given the low Coulombic efficiency of Li plating and stripping in carbonate electrolytes, it seems unlikely that any rest protocol would allow for “complete stripping”, implying that early plating can be completely reversed. If the authors observe this, please explain how this is definitively proven.

Reviewer #2 (Remarks to the Author):

This work investigates the phase-separation and plating phenomenon in graphite-based anode using In operando optical microscope and multiscale phase-field model. The experiments reveal the lithiation and plating/stripping behavior, along with their associated voltage profile for different currents. The study proposes mechanisms for how Li plating occurs at high rates, which are confirmed by multiscale phase-field simulations that aim to correlate microscopic investigations to macroscopic observations. Authors emphasize the importance of relaxation in enabling an efficient fast charging process, and the optimal relaxing times for each rate are suggested. Overall, the manuscript is well written and high quality. However, there are some areas to improve:

1. 3D image-based phase-field model has been used to demonstrate that the early onset of plating is related to a strong dependence of intra-particle lithiation heterogeneity on the particle size, shape, orientation, surface condition and C-rate. Phase-field model relies on the energy functional and thermodynamic/kinetic coefficients. The complexity of the problem should have phase-field model built with some assumptions which however are not clearly provided. In fact, some experimental evidence should be included to justify these assumptions. In other words, it would help if some experimental characterizations on the microscopic plating initiation could be included.
2. Since authors performed deep microscopic investigation, could authors suggest the optimal size and distribution of graphite particles for fast charging applications?
3. The details of how the simulations are performed, including initial/boundary conditions for both the 1D and 3D phase-field models, are not clear.
4. In all figures, the authors should clearly indicate which curve or plot corresponds to experimental observations and which is predicted.

Reviewer #1 (Remarks to the Author):

This article describes a multi-modal study of graphite anodes during charging and OCV rest periods. Gradients in SOC are observed and modeled at both the intra-particle and inter-particle levels. The influence of particle size, shape, and orientation, are discussed on local heterogeneity. Finally, the insertion of 3-min OCV rest periods are studied to allow for SoC relaxation to delay Li plating.

Overall, this is an interesting article that will provide value to the fast-charging community. In particular, the discussion of intra-particle SOC gradients is under-discussed in the literature, and provides high impact. Observations of individual particle shape and orientation effects are interesting and relevant to understand heterogeneity at small length scales. However, there were several confusing and unclear aspects to the discussion, which made the paper hard to follow at times. The reviewer is supportive of publication in Nature Communications, after the following points are addressed:

1) The definition of C-rate and current density during the operando optical microscopy experiments were unclear and difficult to follow at several points in the discussion. First: what area was used to define current density in these experiments? As discussed throughout, the local current density actually varies spatially throughout the electrode surface in thick electrodes. Typically, an areal current density is defined, when the area of the working and counter electrodes are the same, and are placed in a parallel electrode geometry. However, in this optical experiment, it was not clear what direction the electric field is pointing (is it normal to the electrode thickness, or in the lateral direction?) Figure S1 was difficult to interpret, with respect to the location of the working and counter electrodes, and the perspective from which the optical imaging is taken. A more detailed schematic with labels and arrows showing these directions would be helpful.

Based on these difficulties in visualizing the setup, the discussion of “thickness” and “interface” were also confusing, as discussed in a later comment. A significant part of this challenge is that the actual optical cell geometry, with the working, counter electrodes, separator etc. are in the SI. The reviewer feels that this information should be moved to the main text, and the definition of current density and C-rate need to be more clearly explained for the optical cell. For example, the authors state “Here a current density of 2 mA cm⁻² corresponds to the current density of 1C in the coin cell experiment of this study, but the effective C-rate of the former is much lower due to the large geometry of the electrode”. What is meant by “effective C-rate”? Please define current density and C-rate more mathematically for the optical cell, with visualizations of the electric field and electrode orientations in the main text.

The authors would like to express their gratitude to the Reviewer for bringing to their attention the presence of confusing content. The current density was defined as the ratio of the total current and the 4 lateral surface areas of the electrode, as there is no intercalation reaction from the top and bottom planes. The direction of the electric field is pointing in the lateral direction. In response to the Reviewer's request, the authors have made extensive revisions to the manuscript. The first paragraph of the *Results* section now contains more detailed explanations regarding the definition of the current density, the geometry of the cell, and the direction of Li⁺ flow (electric field) in both the lateral and thickness directions. These additions have been highlighted for ease of reference. Furthermore, a new Figure 1 has been included to

provide graphical illustrations of these concepts, replacing the original Figure S1, which has been removed from the Supplementary Information.

With regard to the term "effective C-rate," the authors intended to convey the difficulty in accurately estimating a C-rate in the optical cell geometry. This is due to the large (lateral) thickness of the graphite strip, which results in under-lithiation in the center region caused by the electrolyte concentration gradient. Consequently, the "effective" capacity is actually lower than the nominal capacity. Moreover, despite having an identical areal current density, the C-rate in the optical experiment is significantly smaller compared to the conventional coin cell setup, primarily due to the substantially smaller lateral reacting surface. As a result, the C-rate in the optical experiment cannot be directly compared to that of the coin cell setup. For these reasons, the authors avoided using "C-rate" to interpret the result of the operando optical experiment, instead, the lithiation time in hours is used. To mitigate any confusion stemming from this, the authors have made the decision to remove the term "effective C-rate" from the manuscript. It is important to note that this term appeared only once in the manuscript and its removal has minimal impact on the overall understanding of the content.

2) Following on point 1, how is C-rate defined in the optical experiments? Typically, the C-rate is based on the charge current needed to fully charge the electrode, based on the areal capacity (e.g. 2 mAh/cm²) and corresponding current density. However, it was not clear what the areal capacity is in Fig. 1. The caption lists "the first 400 microns of a total thickness of 2 mm". Was the thickness of the electrode really 2mm? Or is this the "lateral thickness"? If the later, how is C-rate defined? This was confusing.

The authors have made a deliberate choice to exclude the use of the term "C-rate" in reference to the optical experiment, as previously explained in response to the aforementioned comment. As a result, the term "effective-C-rate" has been completely eliminated from the manuscript. To provide further clarification, the total "thickness" refers to the width of the graphite strip, which measures 2.2 mm. This is based on the lithiation direction of the electrode. In order to examine the relevant dynamics, a portion of the full width, specifically the first 400 μm , was observed under a magnification of 700x. This chosen range is sufficiently large to effectively demonstrate the influence of charging current density and electrolyte concentration gradient on the spatial dynamics of lithiation, phase separation, and plating/relaxation.

To aid in better understanding, the revised manuscript includes visual representations of the definition of "thickness" in the form of the new Figure 1, the caption of Figure 2 (previously Figure 1), and the surrounding contextual explanations.

3) The paper discusses "global SOC" several times, but this term is not clearly defined. Is this referring to the charge of the electrode relative to the theoretical capacity of the entire electrode? Please explain how this is used in the various configurations (optical vs. coin cell, etc.).

The Reviewer's is correct, that the global state of charge (SOC) is indeed defined as the charged capacity divided by the theoretical capacity of the electrode. This differs from the particle SOC, which reflects the lithiation state of an individual particle. This was described in the first paragraph under *Results* as "the percentage and decimal number of SOC represents the lithium content in the electrode and particle,

respectively.” The authors have taken the feedback into consideration and have revised the sentence to enhance clarity. The updated sentence now reads as follows: "The percentage and decimal number of SOC represent the lithium content in the entire electrode (i.e., global SOC) and in the particle, respectively."

4) Why was the plated Li in Fig. 1 so out of focus? And in which direction (relative to the microscope and electrode configuration” is the Li plating occurring? Is it constrained to only grow towards the separator (as would be typical in a coin cell), or is it growing out towards the microscope objective? Please clarify this in the text.

The authors would like to provide an explanation regarding the potential factors that may impact the sharpness and clarity of the plated lithium in the images. One possible influence is the formation of gas bubbles around the plated lithium, which can affect its overall appearance. Another factor to consider is the shiny reflection of Li, which can result in the loss of surface detail and affect the visual quality of the images. It is worth noting that these issues are unlikely to be caused by focus-related problems, as all the images were captured using a depth-of-field composition mode. This specific mode was employed to ensure the sharpness and clarity of the images, minimizing any potential focus-related issues.

The growth of plated lithium occurs laterally, specifically at the lateral surface of the electrode (LSE), where it can expand freely without any constraints. Thanks to the Reviewer’s comment, extra labels have been incorporated into Figure 2 to enhance the visual representation and understanding of the lateral growth of plated lithium.

5) The text refers to the EEI as the “electrode electrolyte interface”. Does this mean electrode-separator interface? This terminology is confusing, because there is electrolyte present throughout the porous electrode geometry. Also, is there any electrolyte on the top surface (towards the microscope objective) that is not within the pores of the electrode? If so, wouldn’t there be concentration gradients and electrolyte transport effects within this “top” region (perpendicular to the electrode-separator interface)? This was not clear, and again, might be more clearly shown when adding the cell geometry information to the main text.

The term "EEI" was originally used to refer to the interface between the lateral surface of the electrode and the bulk of the electrolyte. The authors acknowledge that this terminology can be confusing, as the term "EEI" is commonly used to denote the electrode/separator interface in coin cell or pouch cell designs. Therefore, the authors have made the necessary revision in the manuscript. The term "EEI" has been replaced with "LSE" (lateral surface of the electrode) throughout the revised manuscript to provide clarity and avoid any potential confusion.

There is minimal or no presence of electrolyte between the electrode and the glass at the top due to the tightly bonded contact between them. To ensure this intimate contact, any bubbles formed at the interface of the electrode and glass were carefully removed using a syringe during the electrolyte injection into the cell. This procedure further guarantees the establishment of optimal contact. If there were electrolyte present on the top surface of the electrode, one would expect a homogeneous color change of the graphite particles throughout the electrode instead of the observed phenomenon of initiation from the lateral edges during the charging process. In the revised manuscript, the authors have included additional details in the first paragraph of the Results section, as follows “The top surface of the electrode strip is in

close contact with the glass (Fig. 1a), where the electrolyte accessibility is limited, and the bottom of the strip is copper current collector, in contact with the separator. Thus, the Li^+ ions mainly intercalate from the lateral surfaces towards deeper region of the graphite electrode (Fig. 1c and d)."

6) In the paragraph below figure 1, "nominal current density" is used for the first time. What is meant by this, and how does it relate to "effective C-rate" described above?

Nominal current density refers to the intended or expected charging current density to be applied in the experiment, which only lasted for 1 hour due to the occurrence of lithium plating. To prevent any potential confusion, the authors have deleted the word 'nominal'.

7) Overall, the reviewer found the optical microscopy section the most difficult to follow- in the paper. It was not clear how some of these terms were defined (C-rate, current density, etc." relative to the areal capacity fo the porous electrode relative to the current collector, vs. the direction of lithiation. It should be emphasized to the reader if this visualization is actually in the in-plane (lateral direction) of a "typical" electrode, and that the corresponding direction of SoC gradients observed is perpendicular to what is normally considered the "thickness" direction, which was unclear. That would help to clarify several of the points above.

In summary, the authors have made the following changes according to the Reviewer's suggestion:

(1) A new Figure 1 has been introduced, featuring comprehensive labels, arrows, referential axes, and a schematic representation that illustrates the sample geometry, spatial arrangement, viewing angle, and lithiation direction. Notably, the Li^+ insertion direction and the thickness direction have been emphasized with labels and arrows in multiple panels.

(2) Additional descriptions have been added in the first paragraph in *Result* section to provide further clarity and ensure a better understanding of the aforementioned aspects: "The top surface of the electrode strip is in close contact with the glass (Fig. 1a), where the electrolyte accessibility is limited, and the bottom of the strip is copper current collector, in contact with the separator. Thus, the Li^+ ions mainly intercalate from the lateral surfaces towards deeper region of the graphite electrode (Fig. 1c and d), and accordingly the current density is calculated as the ratio of the total current and the four lateral surface areas of the rectangular graphite electrode. Note that the 'thickness' in the optical experiment refers to the in-plane horizontal distance in lithiation direction of the electrode strip (x-direction, Fig. 1c), which is much larger than the conventional thickness of the electrode coating (z-direction, Fig. 1d)."

8) All of the experiments in this paper appear to be 2-electrode measurements with a Li metal counter electrode and no reference electrode. Therefore, in the experimental voltage curves, although the graphite potential is plotted as "vs Li/Li+", the effects of overpotentials at the Li metal counter electrode (which is not at 0V during non-zero currents) and IR drop will impact the measured potential. It was not clear, therefore, how it could be determined that 20 mV is the energy barrier for nucleation. This was also confusing, because it was not clear if the authors were stating that the electric potential of the composite graphite anode (composed of many particles at different SOC) is the nucleation barrier, or if this barrier applies to the electrochemical potential of individual graphite particles, which are at different SOC. Please

clarify how this single number is used in this definition, and how it relates to inter-particle SoC heterogeneity.

The authors acknowledge that the measured voltage may be subject to shifts caused by the overpotential at the Li metal anode and IR drop. However, it is important to clarify that the determination of the plating nucleation barrier of 20 mV was not based on the voltage curve and was not influenced by the IR drop or overpotential at the Li anode (i.e., not by fitting the prediction with the experiment). The nucleation barrier for Li plating is an intrinsic parameter of the material and is associated with the surface condition, including factors such as defects, geometry, crystalline orientation, and the electrolyte used.

Plating initiation occurs at a localised point on the surface of individual graphite particles, rather than occurring uniformly across the electrode. Hence, in this study, the nucleation energy barrier for plating was specifically applied to each individual graphite particle, rather than using a single overpotential value for the entire composite electrode. The plating propensity varies from particle to particle, depending on their unique local electrochemical conditions.

The determination of the nucleation energy barrier was carried out as follows: the plating kinetics (can be found in the SI and it is copied below) indicated that a nucleation energy barrier (E_b) was incorporated into the calculation of the local plating kinetics for each individual particle. Following the initiation of plating, this energy barrier exponentially decreases as the plating thickness increases. This exponential relationship was established based on the findings presented in T. Gao et al.'s research published in *Joule* (5, 393-414) by Bazant and colleagues.

$$J_{ct_plt} = i_{0_plt} \left[\exp\left(\frac{\alpha_{plt}F}{RT} \left(\mu_e - \mu_{p_surf} + E_b \cdot \exp\left(-\frac{L_{plt}}{L_{plt_ref}}\right) \right)\right) - \exp\left(-\frac{(1-\alpha_{plt})F}{RT} \left(\mu_e - \mu_{p_surf} + E_b \cdot \exp\left(-\frac{L_{plt}}{L_{plt_ref}}\right) \right)\right) \right]$$

E_b , the nucleation energy barrier, was determined by comparing the phase and SOC distribution of Li intercalated in the electrode along the horizontal thickness direction. This comparison was made between the experimental observations and predictions (as shown in Figure S4 of the revised Supplementary Information) at the end of the charge and relaxation processes. If E_b is over-estimated, it results in an underestimation of the plated lithium, which in turn leads to an overestimation of the intercalated Li. This is indicated by a shift in the phase boundaries and SOC distribution towards the right-hand side of the field-of-view. Consequently, during the relaxation phase, the propagation of the phase boundary (e.g., the gold/red boundary in Figure S4c and d) resulting from the re-intercalation of the plated lithium is less pronounced due to the underestimated plating. Conversely, if E_b is underestimated, the opposite effects occur. Through careful analysis, it was determined that an E_b value of 20 mV provides the optimal consistency with the experimental results. However, it is worth noting that the predicted relaxation voltage curve exhibits a more extended stripping process compared to the experimental data (as shown by the green curve in Figure S3a). This discrepancy could potentially be attributed to the model's inability to account for the formation of dead lithium during stripping, as it assumes all plated lithium is reversible. Nonetheless, the possible presence of dead Li does not influence our procedure to estimate the nucleation barrier from the SOC distribution and shift in phase boundary during charge.

The authors have made further enhancements to provide additional details regarding the nucleation energy barrier of plating (20 mV) in the relevant sections of the main text (above Fig. 3) as well as in the Supplementary Discussion in the revised SI to address the concerns mentioned above.

9) The discussion of the individual particle geometries, orientations, size effects etc. was great, and a true highlight of the paper. One thing that was not clear is the term “Li flow direction”. What is meant by that? Are the authors referring to the vector pointing normal to the electrode/separator interface? Li is clearly “flowing” in 3-dimensions within this 3-d electrode geometry, so this terminology was not clear. How does this “flow direction” relate to the electric field direction in the electrolyte phase surrounding these individual particles? It seems that the electrolyte potential gradients will also lead to a 3-D distribution of the electric field in the vicinity of the individual particles, so it was not clear what the “flow direction” indicates.

In the main text, the term "Li flow direction" is used to indicate the overall lithiation direction of the electrode. The authors acknowledge that this expression may not be precise, as it could potentially be misconstrued as referring to the local movement of Li ions within the electrolyte. The trajectory of Li ions within the electrolyte is indeed intricate, influenced by factors such as electrolyte concentration, Li ionic potential, and particle geometry.

To eliminate any confusion arising from this terminology, the authors have decided to substitute "Li flow direction" with the phrase "global lithiation direction"

10) The authors state that “intra-particle SOC heterogeneity cannot be captured by macroscopic modelling techniques and yet is the major cause of the early onset of lithium plating”. What is meant by “the major cause”? Inter particle heterogeneity (including SoC gradients throughout the electrode thickness) lead to local current focusing, which accelerates intra-particle heterogeneity. So it seems that both intra- and inter-particle SoC heterogeneity are coupled in a composite electrode, and we can not state that one is the “major cause” vs. the other.

The authors totally agree with the Reviewer’s point of the inter-dependence of the inter-particle and intra-particle SOC heterogeneity. The uneven lithiation at the electrode scale can contribute to intra-particle SOC heterogeneity, which can be influenced by various factors such as surface morphology, shape, and size of the particles as well. Although the authors specifically mentioned intra-particle SOC to emphasize early plating at the particle level due to the heterogeneous SOC distribution, we agree that inter-particle SOC heterogeneity is also significant and should not be overlooked. To address this concern, the authors have incorporated the following change in the revised manuscript: “However, intra and inter-particle SOC heterogeneities cannot be captured by macroscopic modelling techniques and yet are the major causes of the early onset of lithium plating.”

11) The 3-D microstructure characterization section provides details that do not seem to directly relate to the rest of the story of the paper. For example, it was not clear how the details in Fig. 3d-j directly relate to the discussion? Perhaps the most relevant point is the significant differences in tortuosity factor in the vertical vs. horizontal directions. It is worth mentioning that the optical cell geometry will thus experience greater electrolyte transport limitations than the coin cell, if the direction of “thickness” is in the horizontal direction.

The authors acknowledge that the previous manuscript did not adequately illustrate the relationship between the 3-D microstructure characterization and the performance results. In response to this feedback, the revised manuscript now places greater emphasis on the significance of the 3-D microstructure characterization in three key aspects:

1. Parameterization of the graphite electrode in macroscopic modeling: the 3-D microstructure characterization provides valuable input for macroscopic modeling, including parameters such as tortuosity, graphite phase fraction, and porosity. These insights enhance the accuracy and reliability of the modeling approach.
2. Influence of electrolyte concentration gradient: the large electrolyte concentration gradient observed in the study is a direct consequence of the significant Li^+ transport resistance in the pore phase, attributed to factors such as large tortuosity (as illustrated in Figure 4e) and horizontally aligned graphite particles (as depicted in Figure 4f).
3. Local variation of electrolyte concentration: the observed local variation in the pore phase's electrolyte concentration is a result of the uneven spatial distribution of pore sizes, as demonstrated in Figure 4b.

These are embedded in the main text in the revised manuscript. Furthermore, the authors believe that the advanced 3D characterization techniques employed in this study, including the separation of porosity and pore size in both the thickness and in-plane (calendered) directions, offer a comprehensive understanding of the graphite electrode's properties. They anticipate that these findings will benefit researchers and modelers in the battery research community, contributing to advancements in the field.

As the Reviewer points out, the electrolyte transport resistance in the optical experiment is much larger than the coin cell configuration. This discrepancy arises from the substantial thickness in the lithiation direction of the graphite strip employed in the optical experiment. This particular design choice aligns with the primary objective of the optical experiment, which is to capture and analyze the spatial dynamics of lithiation and non-equilibrium phase separation in the presence of an electrolyte concentration gradient. This has been mentioned now at the end of the first paragraph in *Results*.

12) In discussing the OCV profile, the authors state that “The inflection of the OCV curve is known to be an indication of lithium plating^{42, 43}. The curve is divided into two plateaus, the first of which corresponds to the stripping process and the dissolved lithium back-intercalating in a deeper area of the electrode to maintain neutrality, thus slowing down the rise of the OCV curve and delaying the equilibration.”. This plateauing behavior during OCV, and the deflections associated with stripping of plated Li, re-intercalation, and the delayed onset of equilibration are described in detail in Ref. 37, which would be good to cite here. In general, Ref. 37 provides a few relevant observations that further strengthen the discussion in the current paper, including a direct observation that Li plating nucleates on individual particles that lithiate fastest (the particles that experience the highest local SOC and turn gold first are the same particles where Li plating nucleates”), as well as the fact that the OCV peaks shift to longer times when the C-rate and thickness increase (discussion of Fig. 4).

The authors thank the Reviewer for the suggestion. We agree that the experiment results and discussion presented in Ref. 37 significantly enhance the statement and findings of this study. To incorporate these relevant insights, the authors have cited Ref. 37 multiple times throughout the revised manuscript. In addition, we have included additional descriptive texts at specific points in the manuscript that discuss the

mechanism of the inflection point of OCV relaxation, the peak shift of dV/dq , and the relationship between early plating propensity and SOC (above Fig. 5 and Fig. 7 in the revised manuscript).

13) Fig. 4a was difficult to interpret, as the panel is quite small with several labels and multiple voltage curves. It would be helpful to increase the width of this panel in the figure, to improve readability.

As suggested by the Reviewer, the authors have increased the width of Fig. 4a (Fig.5a in the revised manuscript) to improve the readability.

14) It was a bit surprising to see significant Li plating at a 1C rate for relatively “modest” electrode loadings of 1-2 mAh/cm². Do typical commercial Li ion batteries experience plating at 1C for these loadings? It would be useful to benchmark these observations of plating at 1-2C rates for 1-2 mAh/cm² loadings with other reports in the literature; for example K. G. Gallagher et al., J. Electrochem. Soc. 2016, 163 (2), A138–A149.

The authors thank the Reviewer’s comment and agree that it is a good point benchmarking the experiment and modelling with other research. Accordingly, the authors found another publication (W. Cai et al., Angew. Chem. Int. Ed. 2021, 60, 13007) in addition to the suggested paper, both of which reported lithium plating in 2.2 mAh cm⁻² electrode loading at 1C or lower in the commercial pouch cell format. These information and citations have been added and highlighted in the revised manuscript in the paragraph below Fig. 6.

To further substantiate this, the authors have conducted synchrotron X-ray radiography (imaging resolution 0.325 μm) to monitor plating initiation in a 2 mAh cm⁻² graphite electrode charging at 1C (shown below). Compared with the pristine electrode, a thin white line appears at the separator/electrode interface at 60% SOC, due to the formation of lithium plating layer, which has a low density and atomic number, meaning that it is almost transparent (and thus is bright) in the X-ray image. Note that this 1D white line represents the projected plating regions along the beam direction (perpendicular to the screen) from a 2D plating plane (electrode/separator interface). This is consistent with the coin cell experiment (Fig. 5 in the revised manuscript) and prediction by 3D phase-field modelling (Fig. 7 in the revised manuscript). The authors have added discussion of this content below Fig. 6 in the revised manuscript and added this figure as Fig. S9 in the revised SI.

15) No SOC “color bar” was provided in Fig 5a. Is it the same color scale as Fig. 5b? In general, the use of a color scale to represent SoC and current density in the same figure (Fig. 5) makes the figure a bit confusing to follow, especially with two different color scale bars for current density in panels c-e.

The figure the Reviewer refers to is now Fig. 6 in the revised manuscript, so we refer to it in this reply. The color bar for Fig. 6a is the same as Fig. 6b. The authors have added it to Fig. 6a now.

The authors would like to clarify a potential misunderstanding regarding the color scheme used in Fig. 6 as mentioned by the Reviewer. Panels c and d of the figure depict the SOC distribution, utilizing a rainbow color scheme. On the other hand, panel e represents the current density, employing a bi-polar blue-red color scheme. This distinction in color schemes aims to clearly convey the different parameters being visualized in each panel.

Initially, the authors employed the same color scheme for both SOC and current density. However, after careful consideration, we made the decision to modify the color scheme for two primary reasons. Firstly, this alteration was made to enhance the distinguishability between the panels depicting SOC mapping and those illustrating current density. Secondly, and perhaps more significantly, the authors discovered that the utilisation of a bi-polar blue-red color scheme in panel e effectively accentuates the propagation of the reaction front. This approach proved to be more effective in emphasizing the transition compared to a color bar with a smooth gradient across a range of colors.

The color scale bars for the current density (panel e) are different because the magnitude of the current densities charging at 0.05C and 1C are significantly different, thus using the same color scale bar for the two rates will saturate the color distribution in one or the other.

16) In the later parts of the paper, a CC-CV charging protocol with a cutoff voltage in graphite potential was introduced, but this was not clearly explained: was this the first time that a CC-CV protocol was used? And was it only used on some experiments in the paper?

The CC-CV (constant current-constant voltage) charging protocol is widely employed in battery research due to its simplicity of implementation. However, it is not ideal for fast charging applications due to the resulting voltage polarization caused by concentration gradients in both the solid particle and electrolyte phases. This polarization increases with the charging rate and can cause the voltage to reach the cutoff voltage prematurely, leading to capacity loss. To mitigate this, a prolonged constant voltage (CV) step is introduced after the constant current (CC) step to recover the capacity lost during the CC phase due to polarization. This issue becomes even more critical if plating occurs during the CC step, further hindering the relaxation of the concentration gradient.

In the *Introduction*, this limitation of the CC-CV method was highlighted as follows: "*Although easy to implement, the constant current-constant voltage (CC-CV) method is unsuitable for fast charging due to induced high temperature and plating risk.*" Consequently, more advanced protocols have been developed for practical fast charging applications, providing corresponding references [2,32-36]. However, in

laboratory settings, CC-CV with a cutoff voltage remains a conventional protocol for testing the cycling performance and rate capabilities of electrodes.

In this paper, the authors employed the CC-CV protocol to assess the severity of plating and the required equilibration time. This benchmarking was conducted in comparison with the CC-OCV protocol, where a relaxation step follows the CC phase before recharging. By employing these protocols, the authors aimed to evaluate the effects on plating and assess the equilibration dynamics.

17) In the summary of the charge-protocol section, it is stated that the section provides proof that a relaxation step can allow for “complete stripping of any heterogeneous early plating”. It was unclear how this was proven? Given the low Coulombic efficiency of Li plating and stripping in carbonate electrolytes, it seems unlikely that any rest protocol would allow for “complete stripping”, implying that early plating can be completely reversed. If the authors observe this, please explain how this is definitively proven.

The authors would like to provide clarification regarding the term "complete stripping" used in this study. In this context, it refers to the retrieval of reversible (active) plated lithium, excluding the portion of dead lithium detached from the electrode matrix as a result of volume shrinkage. The key point is that by incorporating a short relaxation step at an intermediate state of charge (SOC), as suggested in this study, the extent of early plating can be significantly suppressed. This is attributed to the effective mitigation of the SOC gradient within the particle (Fig. 8). In comparison, a normal charging procedure without the proper relaxation step can lead to more severe plating due to surface saturation, potentially resulting in incomplete stripping of the plated lithium if the relaxation time is insufficient. Additionally, the authors believe that a well-designed relaxation step can also help mitigate capacity loss attributed to dead lithium, particularly when the severity of lithium plating is reduced.

The authors agree that the expression of “complete stripping” is misleading and have rephrased the content in different places as follows: “The results in Fig. 7d-f not only suggest how long the relaxation time should be to complete the stripping of any plated lithium that is active (i.e., in contact with the electrode matrix).....”, “(2) facilitate capacity recovery by a thorough stripping of the plated lithium that is active and reversible; (3) reduce capacity loss arising from the formation of dead lithium;”

Reviewer #2 (Remarks to the Author):

This work investigates the phase-separation and plating phenomenon in graphite-based anode using In operando optical microscope and multiscale phase-field model. The experiments reveal the lithiation and plating/stripping behavior, along with their associated voltage profile for different currents. The study proposes mechanisms for how Li plating occurs at high rates, which are confirmed by multiscale phase-field simulations that aim to correlate microscopic investigations to macroscopic observations. Authors emphasize the importance of relaxation in enabling an efficient fast charging process, and the optimal relaxing times for each rate are suggested. Overall, the manuscript is well written and high quality. However, there are some areas to improve:

1. 3D image-based phase-field model has been used to demonstrate that the early onset of plating is

related to a strong dependence of intra-particle lithiation heterogeneity on the particle size, shape, orientation, surface condition and C-rate. Phase-field model relies on the energy functional and thermodynamic/kinetic coefficients. The complexity of the problem should have phase-field model built with some assumptions which however are not clearly provided. In fact, some experimental evidence should be included to justify these assumptions. In other words, it would help if some experimental characterizations on the microscopic plating initiation could be included.

The Reviewer is correct to point out that phase-field models rely on energy functional and thermodynamic coefficients. In this study, we use a phase-field model for graphite with parameters (semi-empirical) that have been validated against in-situ experiments. These results are outlined in a series of papers led by Bazant, one of the co-authors of this study: *Acc. Chem. Res.* 2013, 46, 5, 1144–1160 as general theory, then more specifically in *J. Phys. Chem. C* 2017, 121, 23, 12505–12523 and *Journal of The Electrochemical Society*, 164 (11) E3291-E3310 (2017), and others. In those papers, the energy functional and the assumptions of the approach are detailed. In this paper, for the sake of brevity, we just cited the seminal papers and report the full list of equations and model description mainly in SI.

The aforementioned papers however only demonstrate the validity of phase-field graphite models under slow charging rates. Under fast charge conditions for a single graphite particle, the validity of the model was demonstrated for the first time in *Joule* 5 (2), 393-414. In our paper, we upgrade that framework for phase-field and Li plating to a multi-particle system. As requested by the Reviewer, the authors have conducted synchrotron X-ray radiography (imaging resolution 0.325 μm) to monitor plating initiation in a 2 mAh cm^{-2} graphite electrode charging at 1C (shown below). Compared with the pristine electrode, a thin white line appears at the separator/electrode interface at 60% SOC, due to the formation of lithium plating layer, which has a low density and atomic number, meaning that it is almost transparent (and thus is bright) in the X-ray image. Note that this 1D white line represents the projected plating regions along the beam direction (perpendicular to the screen) from a 2D plating plane (electrode/separator interface). This is consistent with the coin cell experiment (Fig. 5 in the revised manuscript) and prediction by 3D phase-field modelling (Fig. 7 in the revised manuscript). The authors have added discussion of this content below Fig. 6 in the revised manuscript and added this figure as Fig. S9 in the revised SI.

2. Since authors performed deep microscopic investigation, could authors suggest the optimal size and distribution of graphite particles for fast charging applications?

Unfortunately, providing a simple answer to this question is challenging. The mass transport limitation within the solid phase depends not only on particle size but also on various interconnected geometric and

operational factors, such as electrode thickness, porosity, and C-rate. In other words, there is no universally applicable "optimal" size and distribution of graphite particles that can suit all mass loadings and working conditions.

In the case of high mass loading, the large electrolyte transport resistance can lead to locally high intercalating current density near the electrode/separator interface. The thickness of the electrode (or the applied C-rate) influences the onset of particle saturation in this region. Consequently, smaller particles are required in this area. However, determining the specific size of these particles also depends on the electrode's porosity.

Previous research conducted by the authors suggested that a gradient or layered particle size distribution could be an optimal design strategy for fast charging. However, due to the complexity of the problem, it is not feasible to provide a definitive optimal particle size and distribution. Instead, it is possible to develop a case-specific design solution by considering the specific requirements of the application.

3. The details of how the simulations are performed, including initial/boundary conditions for both the 1D and 3D phase-field models, are not clear.

The model setup and the underlying physics of the macroscopic and 3D phase-field model are explained in the *Method* section. This includes a comprehensive description of the variables, solution domains, and transport and reaction kinetics involved. For a more detailed mathematical understanding of the macroscopic and 3D model, readers can refer to the Supplementary Information. The Supplementary Information provides additional information on the domains and their corresponding governing equations, boundary conditions (Table S2), input parameters (Table S3), and initial conditions. To enhance clarity, the authors have also included additional explanations of the boundary conditions and initial conditions in the Supplementary Information, specifically placed above Table S2.

4. In all figures, the authors should clearly indicate which curve or plot corresponds to experimental observations and which is predicted.

In response to the suggestion from the Reviewer, the authors have included phrases such as "experimentally measured" and "predicted" in the captions of all figures that require annotation for clarification. However, due to the limited space available, directly adding these texts onto the figures would potentially obstruct the essential information we convey.

Peer review comments, further round review

Reviewer #1 (Remarks to the Author):

The authors have provided a detailed and excellent response to the reviewer questions. The reviewer is supportive of publication, with no further revisions needed.

Reviewer #2 (Remarks to the Author):

The authors have made efforts to improve and validate the phase-field model, although the validation is still a rough. Considering the manuscript includes a comprehensive investigation employing operando high-resolution optical microscopy, non-equilibrium thermodynamics and phase-field modelling, I would suggest the acceptance. However, the reviewer still doubts that the phase-field results are qualitative and the quantitative numbers or so-called beautiful images are not credible. More improvements will be needed in the future.

Reviewer #1 (Remarks to the Author):

The authors have provided a detailed and excellent response to the reviewer questions. The reviewer is supportive of publication, with no further revisions needed.

Reviewer #2 (Remarks to the Author):

The authors have made efforts to improve and validate the phase-field model, although the validation is still a rough. Considering the manuscript includes a comprehensive investigation employing operando high-resolution optical microscopy, non-equilibrium thermodynamics and phase-field modelling, I would suggest the acceptance. However, the reviewer still doubts that the phase-field results are qualitative and the quantitative numbers or so-called beautiful images are not credible. More improvements will be needed in the future.

The authors thank the Reviewer's comment. In this study, the phase-field model has undergone comprehensive validation both at a global and local level. The global validation involved comparing voltage vs. charge curves, SOC distribution along the lateral thickness direction, and the onset of plating via voltage relaxation signals. At a local level, validation was performed through analysing the phase separation phenomenon and onset of plating using synchrotron radiography. Encouragingly, the predicted plating distribution at both electrode and particle scales exhibited qualitative agreement with previous experimental observations.

Nevertheless, it is essential to acknowledge that experimentally measuring the plating thickness, determined by thermodynamics and surface reaction kinetics, can be challenging. This is precisely where the phase-field model proves valuable, as it offers the first quantitative estimation of the severity of plating in 3D. This critical insight can significantly contribute to guiding material design and the development of fast charge protocols.

As per the Reviewer's suggestion, the authors are actively addressing this aspect, and future follow-up studies will undoubtedly enhance the validation process, further solidifying the credibility and usefulness of the phase-field model.